# A High Quality Dataset and Reliable Evaluation for Interleaved Image-Text Generation

Yukang Feng[1,2,3]*, Jianwen Sun[1,2,3]*, Chuanhao Li[4], Zizhen Li[1,2,3], Jiaxin Ai[2,4],
Fanrui Zhang[2], Sizhuo Zhou[2], Yifan Chang[2], Shenglin Zhang[1], Yu Dai[1], Kaipeng Zhang[2,3]†
[1]Nankai University [2]Shanghai Innovation Institute [3]Shanda AI Research [4]Shanghai AI Laboratory
yukangfeng@mail.nankai.edu.cn kaipeng.zhang@shanda.com

## Abstract

Recent advancements in Large Multimodal Models (LMMs) have significantly improved multimodal understanding and generation. However, these models still struggle to generate tightly interleaved image-text outputs, primarily due to the limited scale, quality, and instructional richness of current training datasets. To address this, we introduce **InterSyn**, a dataset that features: (1) large scale, comprising 1.8M multimodal samples; (2) high quality, supported by our proposed **Self-Evaluation with Iterative Refinement (SEIR)** method for rigorous automated quality refinement; (3) rich instructional diversity, ensured through diverse well-designed question templates, based on human preferences and covering a 3500-topic hierarchy. These characteristics make InterSyn particularly well-suited for training LMMs in interactive image–text generation capabilities. To evaluate the capabilities, we propose **SynJudge**, a reliable automatic evaluator that aligns closely with human judge and outputs four interpretable scores: Text Content Completeness (TCC), Image Content Completeness (ICC), Image Quality (IQ), and Image–Text Synergy (ITS). These scores are complementary, covering both content and quality as well as cross-modal interaction, thereby forming a comprehensive evaluation framework. Experimental results on InterSyn subsets of up to 200K samples show that 25K–50K already yield substantial improvements, while scaling to 100K/200K brings further gains in TCC, ICC, and especially ITS, highlighting InterSyn's: (1) scalability, as performance consistently improves with more data; (2) efficiency, as significant gains are achievable even with smaller subsets, making it accessible to researchers with varying computational resources.

## 1 Introduction

Multimodal understanding and generation are critical capabilities toward artificial general intelligence. In the past two years, multimodal large language models (MLLMs) (Liu et al., 2023; Chen et al., 2024c; Wang et al., 2024a) have shown remarkable performance in multimodal understanding and even surpassed humans in some areas, while we have also seen many impressive advances in high quality image generation (Esser et al., 2024b; Betker et al., 2023). However, these models are often limited to generating either text or image outputs in isolation, while real-world scenarios typically require tightly interleaved multimodal outputs.

Recently, pioneer unified LMMs, such as Janus-Pro (Chen et al., 2025b), have shown great potential. However, they struggle to generate instruction-following interleaved image-text outputs, manifesting issues such as semantic drift, low image–text synergy, and poor image quality.

The main challenges lie in the limited scale, quality, and instructional richness of existing datasets. Even with existing datasets (Zhu et al., 2023; Laurençon et al., 2023; Chen et al., 2024a;b; Xu et al., 2024), these challenges remain due to their critical limitations: (1) **Limited scale:** Focus on narrow tasks and typically contain no more than tens of thousands of samples, limiting their applicability to broader real-world scenarios; (2) **Unstable quality:** Built on web-crawled sources (Yang et al.,

---

[1]* Equal contributions.
[2]† Corresponding author.

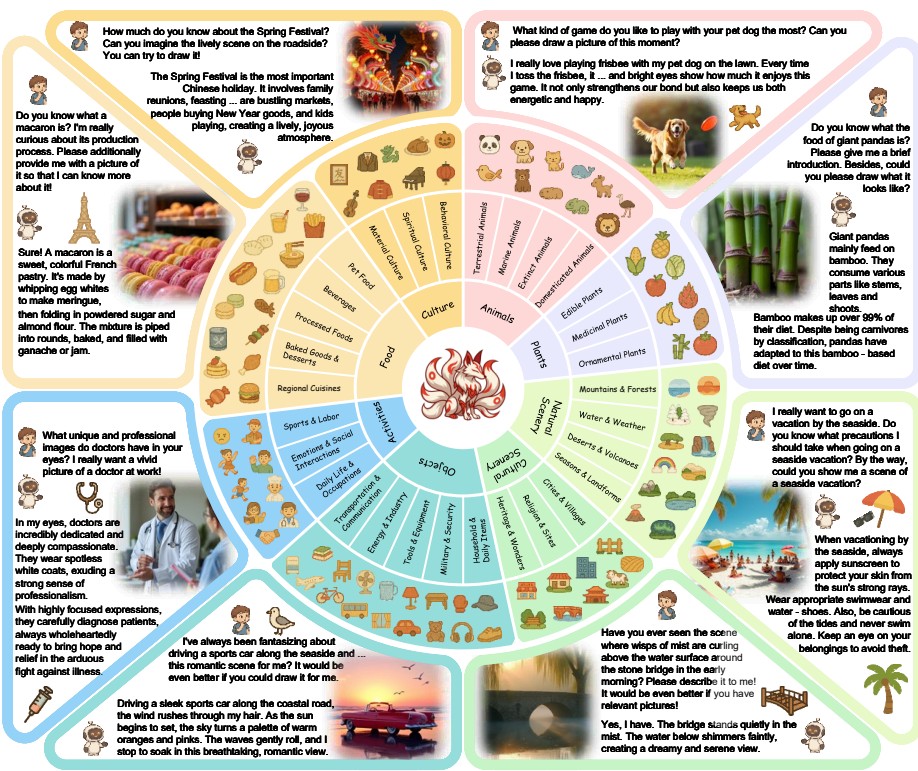

Figure 1: InterSyn: Topic hierarchy and interleaved question answering samples

2021; Laurençon et al., 2023) or reused corpora (Zhou et al., 2018; Zhu et al., 2023) with inconsistent quality and lack standardized quality control mechanisms; (3) **Low interaction complexity:** Rely on static documents or single-turn prompts, thus failing to capture the contextual richness and interleaved structure of authentic human conversations.

To address the above issues, we introduce **InterSyn**—the first fully automated, high quality, large scale dataset for instruction-following, multi-turn question answering with interleaved image–text responses. With 1.8M single-turn and 50,000 multi-turn dialogues across 8 domains and 3,500 topics (as shown in Figure 1), InterSyn provides extensive coverage for diverse real-world scenarios. InterSyn's quality is further enhanced by our **Self-Evaluation with Iterative Refinement (SEIR)** method, which embeds self-checking and feedback loops into each generation step, enhancing semantic completeness and cross-modal synergy. To ensure rich instructional diversity, we extract human-like query styles through question templates that capture varied linguistic structures, supported by a rich topic hierarchy for diverse, instruction-driven dialogues.

To evaluate interleaved image–text generation, several related benchmarks (An et al., 2024; Zhou et al., 2024; Chen et al., 2025a; Xia et al., 2024; Liu et al., 2024) have been proposed, but they still suffer from the following limitations: (1) **Limited domain and scale:** small, task-specific sets cannot cover realistic multi-turn dialogue needs; (2) **Costly manual evaluation:** accurate assessment still hinges on human, whose expense and delay hinder large scale, rapid benchmarking; (3) **Weak alignment with human preference:** current automatic metrics significantly diverge from human judgments on fine-grained multimodal reasoning; (4) **Narrow evaluation scope:** emphasise surface correctness while overlooking synergy and overall answer quality.

Therefore, we propose **SynJudge**, a reliable and comprehensive judge model for evaluating interleaved image-text generation with high alignment to human judgment. SynJudge provides interpretable, quantitative feedback across four key dimensions: text content completeness (TCC),

image content completeness (ICC), image quality (IQ), and image-text synergy (ITS). Unlike traditional image–text consistency metrics, the ITS metric focuses on rewarding tight, complementary alignment between the textual and visual modalities while penalizing redundancy.

To validate our contributions, we conduct a series of experiments. First, the effectiveness of our SEIR method is demonstrated, showing substantial quality improvements over a non-refined baseline. Additionally, comparisons with several existing models show that SEIR consistently outperforms them across all evaluation metrics. Next, we validate SynJudge, which exhibits the strongest alignment with human judgment, showing a significantly smaller deviation compared to zero-shot MLLM evaluators. Finally, we validate the utility of InterSyn by fine-tuning models on randomly sampled subsets, up to 200K examples. Results reveal significant performance gains with as few as 25K/50K samples, underscoring the dataset's high data density and efficiency. These results confirm that our methods and dataset contribute to improved multimodal model performance.

Our main contributions are summarized as follows: (1) We present **InterSyn**, a large scale dataset of 1.8M high-fidelity samples, distinguished by its instructionally rich and complex dialogues that span over 3,500 topics; (2) We propose **SEIR**, a method that ensures high quality data generation across refinement steps with minimal manual effort; (3) We introduce **SynJudge**, a multi-dimensional evaluation model for scoring interleaved outputs, enabling fine-grained assessment and effective feedback for model improvement; (4) We conduct comprehensive experiments demonstrating that InterSyn substantially enhances LMM performance in instruction alignment, image-text synergy, and multi-turn reasoning, contributing to the advancement of unified multimodal systems.

## 2  RELATED WORK

**Models for Interleaved Image-Text Generation.**  Recent advances in MLLMs, such as Flamingo (Alayrac et al., 2022), InternVL (Chen et al., 2024d), and Qwen-VL (Wang et al., 2024a), have substantially improved multimodal understanding. Meanwhile, diffusion models (Ramesh et al., 2022; Betker et al., 2023; Esser et al., 2024a) achieve strong visual generation performance. To unify understanding and generation, models such as MiniGPT-5 (Zheng et al., 2023) and Show-o (Xie et al., 2024) combine autoregressive text generation with diffusion-based image synthesis. More recent efforts  (Team, 2024; Wu et al., 2024; Wang et al., 2024b; Chern et al., 2024) adopt unified autoregressive frameworks for interleaved generation. However, lacking targeted, high quality training data, these models are not explicitly optimized for instruction-following and often struggle to maintain coherence and cross-modal consistency—a gap InterSyn is specifically designed to fill.

**Datasets for Interleaved Image-Text Generation.** High quality interleaved image-text data is crucial for training multimodal models. Existing large scale datasets like MMC4 (Zhu et al., 2023), OBELICS (Laurençon et al., 2023), and CoMM (Chen et al., 2024b) are primarily document-level corpora constructed from web sources, but often suffer from noise, weak alignment and low interaction intensity. Several benchmarks, such as OpenLEAF (An et al., 2024), InterleavedBench (Liu et al., 2024), and OpenING (Zhou et al., 2024), focus on specific tasks. LeafInstruct (Xu et al., 2024) constructs an interleaved image-text dataset by filtering samples from existing corpora (Zhu et al., 2023; Huang et al., 2016; Zhou et al., 2018). However, both benchmarks and datasets remain limited in scale and instructional diversity. To this end, we introduce InterSyn, a large scale, high quality dataset with diverse, multi-turn dialogues and automated refinement.

**Evaluation for Interleaved Image-Text Outputs** Early multimodal evaluation metrics independently assessed text quality (Papineni et al., 2002; Lin, 2004) and image quality (Heusel et al., 2017; Salimans et al., 2016). Subsequent metrics (Hessel et al., 2021; Li et al., 2023; Lin et al., 2024; Chen et al., 2023; Lu et al., 2024) targeted image-text consistency, yet still inadequately evaluated the quality of interleaved outputs. More recent efforts, including InterleavedEval (Xu et al., 2024) and CoMM (Chen et al., 2024b), leveraged MLLMs for holistic assessment, but often exhibit misalignment with human judgment. OpenING (Zhou et al., 2024) proposed IntJudge for pairwise comparisons, but it lacks fine-grained, quantitative scoring for individual responses, limiting its applicability for model training and refinement. In contrast, the proposed SynJudge provides a more comprehensive evaluation by assessing both content completeness and modality synergy, aligning more closely with human judgment.

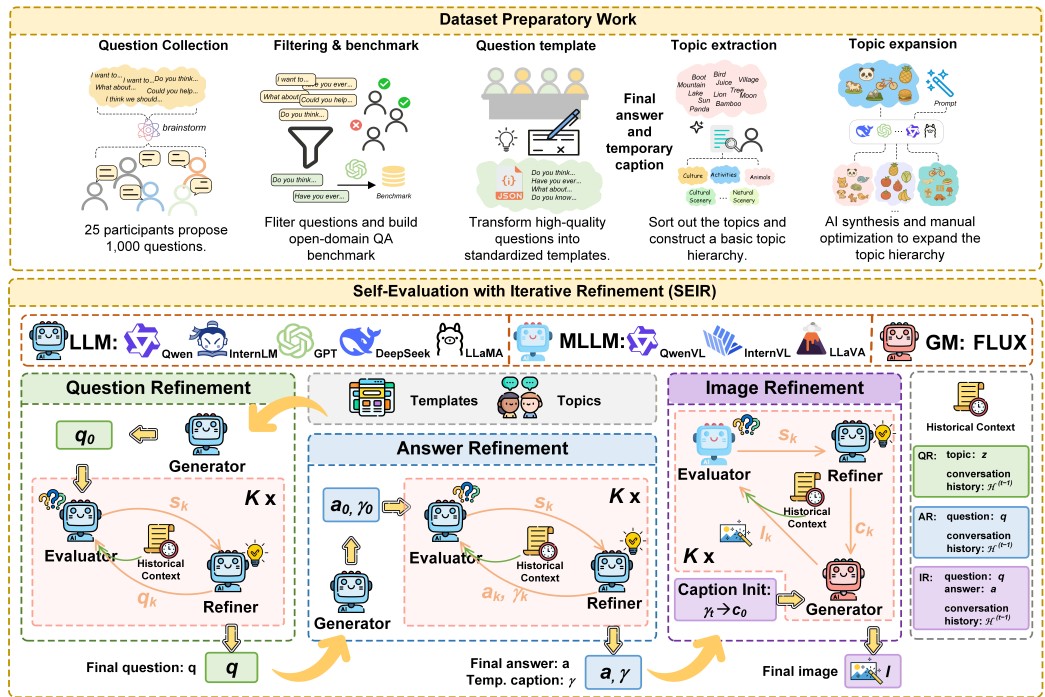

Figure 2: **Overview of the InterSyn Dataset Construction Framework.** The top panel illustrates the dataset preparatory work, covering question collection, filtering, template standardization, and topic expansion. The bottom panel illustrates the **Self-Evaluation with Iterative Refinement (SEIR)** method, which employs a *Generate-Evaluate-Refine* loop across three cascaded stages. (1) Question Refinement (QR): An initial question $q_0$ is refined into the final question $q$ based on the topic $z$ and question template. (2) Answer Refinement (AR): Using $q$, the final answer $a$ and a temporary caption $\gamma$ are iteratively refined. (3) Image Refinement (IR): Initialize $c_0$ with $\gamma$. Refine the caption and image until the final image $I$ is produced. The right-side legend details the inputs and historical context ($\mathcal{H}^{(t-1)}$) used at each stage.

# 3 INTERSYN DATASET AND SYNJUDGE EVALUATOR

## 3.1 OVERVIEW

In this section, we present a comprehensive framework for both the construction and evaluation of interleaved image-text generation. First, we describe the InterSyn dataset construction pipeline, detailing the preparatory work and our proposed **SEIR** method (illustrated in Figure 2). Subsequently, we introduce **SynJudge**, a specialized evaluator designed to assess the quality of interleaved outputs, along with the specific metrics and benchmarks established to validate the model's performance.

## 3.2 DATASET PREPARATORY WORK

InterSyn's preparatory work involves five major stages:

**Question Collection.** We recruited 25 participants, each providing 40 questions drawn from natural conversational scenarios, resulting in a total of 1,000 questions.

**Question Filtering and Benchmark.** We combined LLM-based filtering and expert review to select high quality questions. Redundant, ambiguous, uncommon, and overly subjective samples were removed based on predefined criteria. In total, 500 questions were reserved as a **basic** benchmark.

**Question Template Extraction.** Based on the selected high quality questions, we constructed a set of generalized question templates that capture the conversational query style (i.e., the common lin-

guistic forms people use to pose requests), independent of any specific knowledge content, thereby enabling scalable question generation. See the appendix §E for details.

**Basic Topic Hierarchy.** We performed AI-assisted topic extraction from the filtered questions and manually organized the results to build a basic topic hierarchy, ensuring clear logical dependencies and coherent topic relations.

**Topic Hierarchy Expansion.** We further refined and expanded the basic topic hierarchy to improve both coverage and granularity. Combining AI-assisted topic suggestions with expert curation, we constructed a well-structured hierarchy that supports diverse and scalable data generation.

## 3.3 SEIR METHOD

The Self-Evaluation with Iterative Refinement (SEIR) method enhances data quality via a *Generate-Evaluate-Refine* loop. **For each conversation turn** $t$, we define a **generic refinement operator** $\Phi$ that transforms an **initial content** (e.g., a question or answer) $x_0$ into a **final content** $x$ through $K$ iterations. The loop terminates when the evaluator returns no suggestions or when a maximum iteration depth $K$ is reached. The update rule at iteration $k$ is:

$$x_k = \mathcal{M}_{refine}(x_{k-1}, s_k), \quad \text{where } s_k = \mathcal{M}_{eval}(x_{k-1}, \mathcal{C}). \tag{1}$$

Here, $\mathcal{M}_{eval}$ acts as a judge, analyzing the current content $x_{k-1}$ against context $\mathcal{C}$ (e.g., the target topic, preceding question, or dialogue history) to provide specific suggestions $s_k$. $\mathcal{M}_{refine}$ then produces an improved version $x_k$. We apply this operator sequentially across three stages for the current turn $t$:

**Step 1: Question Refinement (QR).** We first generate an **initial question** $q_0^{(t)}$ from a template and topic. To ensure clarity and depth, we apply $\Phi$ to obtain the **final question** $q^{(t)}$:

$$q^{(t)} = \Phi(q_0^{(t)} \mid \mathcal{C} = \{z, \mathcal{H}^{(t-1)}\}). \tag{2}$$

The evaluator critiques the initial question $q_0^{(t)}$ based on the topic $z$ and conversation history $\mathcal{H}^{(t-1)}$, continuing the refinement until the question meets quality standards or the iteration limit is reached.

**Step 2: Answer Refinement (AR).** Using the final question $q^{(t)}$, we generate an **initial answer-caption pair** $(a_0^{(t)}, \gamma_0^{(t)})$. The refinement loop ensures the text is comprehensive and the caption is relevant, yielding the **final answer** $a^{(t)}$ and a **temporary caption** $\gamma^{(t)}$:

$$(a^{(t)}, \gamma^{(t)}) = \Phi((a_0^{(t)}, \gamma_0^{(t)}) \mid \mathcal{C} = \{q^{(t)}, \mathcal{H}^{(t-1)}\}). \tag{3}$$

**Step 3: Image Refinement (IR).** Finally, we generate and refine the image. We utilize the temporary caption $\gamma^{(t)}$ from AR as the **initial caption** $c_0^{(t)}$. In each iteration $k$, we generate an image $I_k^{(t)}$, which is then critiqued by a VLM ($\mathcal{V}$) against the textual context $(q^{(t)}, a^{(t)})$ and dialogue history $\mathcal{H}^{(t-1)}$. The VLM's feedback $s_c^{(k)}$ guides the generation of a **refined caption** $c_{k+1}^{(t)}$:

$$I_k^{(t)} = \mathcal{G}(c_k^{(t)}), \quad s_c^{(k)} = \mathcal{V}(I_k^{(t)}, q^{(t)}, a^{(t)}, \mathcal{H}^{(t-1)}), \quad c_{k+1}^{(t)} = \mathcal{M}_{refine}(c_k^{(t)}, s_c^{(k)}). \tag{4}$$

This cycle repeats until satisfaction or the iteration limit $K$ is met, yielding the **final image** $I^{(t)}$.

## 3.4 INTERSYN COMPOSITION

InterSyn contains approximately 1.8 million single-turn samples and 50,000 multi-turn dialogues. Data quality is ensured through the SEIR method, which iteratively refines samples to improve answer accuracy, conversational coherence, and image-text synergy. InterSyn offers a rare combination of diversity and quality, providing a robust foundation for training unified multimodal models with strong instruction-following and contextual reasoning capabilities.

### 3.5 SYNJUDGE: A RELIABLE EVALUATOR FOR INTERLEAVED OUTPUTS

To reliably evaluate the complex, instruction-following capabilities of interleaved image-text generators, we propose **SynJudge**, a reliable and comprehensive judge model that demonstrates high alignment with human judgment.

#### 3.5.1 EVALUATION DIMENSIONS

SynJudge is designed to provide interpretable, quantitative feedback by assessing responses across four key, complementary dimensions (See the appendix §F for details.):

**Text Content Completeness (TCC):** Assesses whether the generated text completely and accurately answers the user's question, covering all sub-tasks and constraints.

**Image Content Completeness (ICC):** Assesses whether the generated images correctly depict the key objects, scenes, or concepts requested in the prompt.

**Image Quality (IQ):** Evaluates the visual fidelity, aesthetics, and overall quality of the generated images, penalizing artifacts, distortions, or low resolution.

**Image–Text Synergy (ITS):** Measures the cross-modal relationship. Unlike simple consistency metrics, ITS specifically rewards tight, complementary alignment where the text and images work together to form a cohesive answer. It penalizes redundancy (e.g., text merely describing the image) and irrelevance.

#### 3.5.2 TRAINING AND MODEL SELECTION

To ensure robust evaluation, we first constructed a high-quality, human-annotated dataset. The process began by collecting a diverse set of interleaved responses from various generators (detailed in §4.1.2). These responses were then rigorously scored by a panel of ten expert annotators across the four dimensions (TCC, ICC, IQ, ITS). To ensure high reliability, each sample was scored by multiple experts. Any scoring inconsistencies were resolved through a mandatory discussion protocol until a unified, final score was reached for each sample. This effort yielded a total of 48,000 annotated pairs, which were split into a **38,400-sample training set** and a held-out **9,600-sample validation set**. Using this data, we fine-tuned strong MLLM backbones, including QwenVL and InternVL. As demonstrated in our experiments (see §4.4), the model trained from **QwenVL** achieved the strongest alignment with human judgment, exhibiting the lowest RMSE and highest agreement (A@1). Consequently, we designate this **QwenVL-trained** model as our final **SynJudge**.

#### 3.5.3 EVALUATION BENCHMARK FOR GENERATORS

To rigorously assess the quality of interleaved image-text outputs, we established a fixed **Evaluation Benchmark**. This benchmark comprises a curated series of **4,000** questions (500 human-authored from §3.2 and 3,500 SEIR-generated) spanning the full topic hierarchy. By requiring different models to generate responses to this identical set of questions, we ensure a fair, standardized, and comprehensive comparison of their instruction-following and multimodal generation capabilities.

## 4 EXPERIMENT

### 4.1 EXPERIMENTAL SETUP

#### 4.1.1 EVALUATION DIMENSIONS FOR INTERLEAVED OUTPUTS

We adopt the four evaluation dimensions defined in §3.5.1: TCC, ICC, IQ, and ITS. Briefly, TCC and ICC assess content correctness in text and image respectively; IQ focuses on visual fidelity; and ITS specifically measures the cross-modal synergy and alignment.

To quantitatively assess generators' performance, we adopt the **mean score** (average score across every dimension) and **variance score** (stability across questions). See the appendix §D.2 for details.

### 4.1.2 Evaluated Generators

We evaluate 13 multimodal generators $G$ capable of producing interleaved image-text outputs and we categorize them into two groups based on whether they natively support interleaved generation:

(1) **Non-Interleaved Generators**: These generators produce text and images sequentially through modular pipelines, which include: Emu3 (Wang et al., 2024b), Janus-Pro (Chen et al., 2025b), VILA-U (Wu et al., 2024), Show-o (Xie et al., 2024), Show-o-Turbo (Xu et al., 2025), Liquid (Wu et al., 2025), D-DiT (Li et al., 2025), GPT-4o (OpenAI, 2024) + DALL-E3 (Betker et al., 2023), and Gemini2.5 (Comanici et al., 2025) + FLUX (Black Forest Labs, 2024).

(2) **Interleaved Generators**: These generators can generate interleaved image-text outputs within a unified process, including VARGPT (Zhuang et al., 2025a), VARGPT-v1.1 (Zhuang et al., 2025b), Anole (Chern et al., 2024) and BAGEL (Deng et al., 2025).

## 4.2 Efficiency of InterSyn

### 4.2.1 Data Efficiency and Scalability

We fine-tune four generators on randomly sampled InterSyn subsets of sizes 25k/50k/100k/200k and evaluate with SynJudge. Table 1 shows consistent improvements as data increases. Notably, even **25k/50k** samples already yield clear gains across all dimensions, and further scaling to **200k** continues to improve TCC, ICC, and especially ITS, which highlights the InterSyn dataset's effectiveness in enhancing both semantic alignment and answer completeness.

Table 1: Fine-tuning results on varying subset sizes of InterSyn. Performance consistently improves as training data scales from 25K to 200K samples, demonstrating the dataset's effectiveness and scalability. Notably, just 50K samples yield substantial gains across all models, with continued improvement in content and synergy metrics (TCC, ICC, ITS) at larger scales. All scores are SynJudge means.

| Subset | Anole | | | | VILA-U | | | | VARGPT-v1.1 | | | | BAGEL | | | |
|---|---|---|---|---|---|---|---|---|---|---|---|---|---|---|---|---|
| | TCC | ICC | IQ | ITS | TCC | ICC | IQ | ITS | TCC | ICC | IQ | ITS | TCC | ICC | IQ | ITS |
| baseline | 3.09 | 3.01 | 2.92 | 2.26 | 2.46 | 3.72 | 3.37 | 2.19 | 3.26 | 1.01 | 1.23 | 0.68 | 3.11 | 3.89 | 4.23 | 2.87 |
| + 25k | 3.35 | 3.25 | 3.01 | 2.40 | 2.95 | 3.78 | 3.38 | 2.90 | 3.51 | 2.45 | 2.90 | 2.55 | 3.45 | 4.02 | 4.21 | 3.28 |
| + 50k | 3.47 | 3.28 | 3.10 | 2.74 | 3.19 | 3.83 | 3.39 | 3.20 | 3.68 | 3.12 | 3.67 | 3.00 | 3.69 | 4.11 | 4.19 | 3.56 |
| + 100k | 3.51 | 3.41 | 3.13 | 2.87 | 3.33 | 3.88 | 3.31 | 3.28 | 3.73 | 3.22 | 3.66 | 3.20 | 3.87 | 4.09 | 4.31 | 3.78 |
| + 200k | 3.64 | 3.52 | 3.08 | 3.11 | 3.49 | 4.01 | 3.40 | 3.47 | 3.86 | 3.39 | 3.72 | 3.53 | 4.13 | 4.18 | 4.25 | 4.02 |

### 4.2.2 Retention of General Understanding

Table 2: Understanding performance after 50k InterSyn fine-tuning. Values in parentheses denote the change ($\Delta$) from the base.

| Model | MME-P | MMBench | MMMU | SEEDBench |
|---|---|---|---|---|
| VILA-U | 1344 (+8) | – | – | 57.1 (+0.8) |
| VARGPT | 1465 (-23) | 66.8 (-0.8) | 37.2 (+0.8) | 65.6 (-2.3) |
| VARGPT-v1.1 | 1658 (-26) | 79.4 (-1.6) | 46.2 (-2.3) | 75.2 (-0.9) |
| BAGEL | 1646 (-41) | 83.1 (-1.9) | 52.8 (-2.5) | – |

Crucially, the substantial gains in interleaved generation capabilities (demonstrated in §4.2.1) do not come at the cost of core understanding performance. As shown in Table 2, the models' performance on standard understanding benchmarks (Fu et al., 2023; Liu et al., 2025; Yue et al., 2024; Li et al., 2024) remains robust after fine-tuning.

### 4.2.3 Effectiveness of Multi-turn Data

To verify the effectiveness of our multi-turn dialogue data, we designed an experiment to assess its impact on a models' conversational capabilities. We fine-tuned two models capable of multi-turn generation, Anole and VARGPT-v1.1, on different compositions of single-turn and multi-turn data, while keeping the total training size fixed at 50k samples. The goal is to demonstrate that training with multi-turn data enhances a model's ability to maintain context and quality across an extended conversation. The performance, evaluated by TCC, ICC, IQ, and ITS, is presented in Table 3.

The results clearly demonstrate the value of multi-turn training data. First, the inclusion of multi-turn data does not compromise first-turn performance, which remains high across all trained settings. This is expected, as the SEIR pipeline ensures comparable data quality regardless of dialogue type.

Table 3: Effectiveness of multi-turn data on conversational performance across across different dialogue test turns. Models are trained on different proportions of single-turn and multi-turn data.

| Model | Setting | Training Data | | Turn 1 | | | | Turn 2 | | | | Turn 3 | | | |
|---|---|---|---|---|---|---|---|---|---|---|---|---|---|---|---|
| | | Single | Multi | TCC | ICC | IQ | ITS | TCC | ICC | IQ | ITS | TCC | ICC | IQ | ITS |
| Anole | Baseline | - | - | 3.09 | 3.01 | 2.92 | 2.26 | 2.80 | 2.75 | 2.60 | 1.90 | 2.40 | 2.30 | 2.10 | 1.40 |
| | Trained | 50k | 0 | 3.47 | 3.28 | 3.10 | 2.74 | 3.25 | 2.85 | 2.70 | 2.40 | 2.85 | 2.60 | 2.45 | 2.05 |
| | Trained | 25k | 25k | 3.48 | 3.27 | 3.13 | 2.77 | 3.35 | 3.00 | 2.80 | 2.40 | 3.00 | 2.70 | 2.55 | 2.10 |
| | Trained | 0 | 50k | 3.52 | 3.24 | 3.10 | 2.94 | 3.40 | 3.05 | 2.90 | 2.55 | 3.27 | 2.85 | 2.70 | 2.25 |
| VARGPT-v1.1 | Baseline | - | - | 3.26 | 1.01 | 1.23 | 0.68 | 3.10 | 0.95 | 1.18 | 0.72 | 2.90 | 0.97 | 0.90 | 0.65 |
| | Trained | 50k | 0 | 3.68 | 3.12 | 3.67 | 3.00 | 3.40 | 2.90 | 3.45 | 2.90 | 3.05 | 2.60 | 3.05 | 2.65 |
| | Trained | 25k | 25k | 3.65 | 3.18 | 3.66 | 3.18 | 3.45 | 2.95 | 3.40 | 2.90 | 3.21 | 2.78 | 3.24 | 2.66 |
| | Trained | 0 | 50k | 3.64 | 3.20 | 3.68 | 3.11 | 3.58 | 3.10 | 3.52 | 3.05 | 3.48 | 2.95 | 3.45 | 2.90 |

Second, and more importantly, multi-turn training significantly reduces performance degradation in later turns. Models trained only on single-turn data show a steeper drop in quality as the conversation progresses. In contrast, training with multi-turn dialogues mitigates this degradation, particularly for ITS, by explicitly teaching the model to maintain context across extended conversational dependencies. This confirms the effectiveness of our multi-turn dataset in fostering more coherent and consistent multi-turn generation.

## 4.3 EFFECTIVENESS OF SEIR

To validate the effectiveness of the SEIR, we conduct experiments on both the iterative refinement process and the final output quality comparison with other generators.

### 4.3.1 VALIDATION OF ITERATION REFINEMENT

In this section, all evaluations in this section are conducted by human judge to establish a ground truth. For question quality, Figure 3 shows that QR improves quality over the first three iterations but plateaus thereafter. Based on this, we set the QR to 3, achieving 99.5% of peak quality while reducing computational cost by 40%. For answer quality, we evaluate across different iterations of AR and IR, using a set of 7,000 questions generated through three rounds of QR. As shown in Table 4, the results confirm that AR primarily improves content completeness, while IR enhances multimodal synergy, demonstrating the effectiveness of the SEIR method. Experimental results show that when both AR and IR are set to 4 or 5, the improvements become marginal. Based on this, we fix the number of AR and IR iterations to 3 in dataset construction settings.

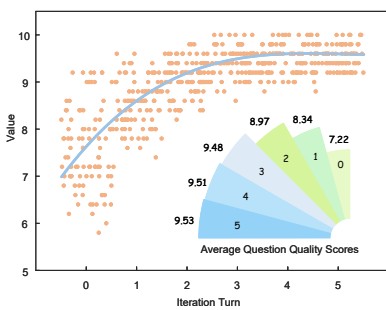

Figure 3: **Impact of question refinement (QR) on question quality.** This plot shows the quality scores across different QR iterations. Quality improves significantly over the first three iterations but plateaus thereafter.

Table 4: Impact of answer refinement (AR) and image refinement (IR) on answer quality. The table reports human evaluation **mean** scores across four dimensions (TCC, ICC, IQ, ITS). AR improves TCC, ICC, and ITS, while IR further enhances ICC and ITS, confirming the effectiveness of iterative refinement.

| AR | IR | TCC | ICC | IQ | ITS |
|---|---|---|---|---|---|
| 0 | 0 | 3.85 | 4.01 | 4.42 | 3.79 |
| 1 | 1 | 4.19 | 4.21 | 4.38 | 4.07 |
| 2 | 2 | 4.34 | 4.35 | 4.41 | 4.38 |
| 3 | 0 | 4.42 | 4.11 | 4.37 | 4.04 |
| 3 | 1 | 4.42 | 4.33 | 4.43 | 4.35 |
| 3 | 2 | 4.42 | 4.43 | 4.39 | 4.46 |
| 3 | 3 | 4.42 | 4.47 | 4.44 | 4.52 |
| 4 | 4 | 4.44 | 4.50 | 4.45 | 4.53 |
| 5 | 5 | 4.45 | 4.49 | 4.43 | 4.53 |

### 4.3.2 COMPARISON OF SEIR OUTPUTS WITH OTHER GENERATORS

We evaluate the SEIR pipeline against 13 baseline generators using the fixed **Evaluation Benchmark** defined in §3.5.3. In this section, we employ both human judges and SynJudge to assess the generator outputs across the four evaluation dimensions.

As shown in Table 5, both human judges and SynJudge confirm that SEIR-generated samples (InterSyn) achieve the highest mean scores across all dimensions. They outperform the strongest baseline, GPT-4o+DALL-E, by a margin of 0.34–0.66, with the largest gap observed in ITS. Furthermore, InterSyn exhibits very low variance (below 0.61), reflecting its consistent output quality and the robustness of our automated method. Crucially, the significant performance gap between InterSyn and the best SOTA generators reveals that even top models still struggle with image–text alignment and complementarity, indicating substantial room for future improvement.

Table 5: Generator performance evaluated by human judge and SynJudge. Each entry is reported as mean (variance), where the value outside the parentheses denotes the mean score and the value inside the parentheses denotes the variance.

| Generator | Human | | | | SynJudge | | | |
|---|---|---|---|---|---|---|---|---|
| | TCC | ICC | IQ | ITS | TCC | ICC | IQ | ITS |
| Anole | 3.06 (1.47) | 2.95 (2.05) | 2.89 (1.87) | 2.25 (2.47) | 3.09 (1.47) | 3.01 (2.11) | 2.92 (1.93) | 2.26 (2.55) |
| DDiT | 0.38 (1.17) | 3.51 (0.83) | 3.29 (0.85) | 0.37 (1.21) | 0.28 (0.86) | 3.67 (0.75) | 3.34 (0.92) | 0.26 (0.87) |
| Emu3 | 3.38 (0.86) | 3.86 (0.86) | 3.87 (0.59) | 3.26 (1.31) | 3.37 (0.74) | 3.92 (0.66) | 3.85 (0.80) | 3.37 (1.16) |
| VILA-U | 2.45 (2.58) | 3.62 (0.91) | 3.35 (0.97) | 2.31 (3.01) | 2.46 (2.73) | 3.72 (0.92) | 3.37 (1.12) | 2.19 (2.96) |
| Liquid | 2.88 (0.76) | 3.82 (0.69) | 3.67 (0.70) | 3.00 (1.42) | 2.87 (0.71) | 3.86 (0.75) | 3.76 (0.85) | 3.06 (1.36) |
| Janus-Pro | 2.93 (0.95) | 3.25 (1.21) | 3.16 (0.99) | 2.55 (1.75) | 2.96 (0.96) | 3.30 (1.09) | 3.11 (1.11) | 2.62 (1.87) |
| Show-o | 3.32 (1.31) | 3.65 (0.94) | 3.52 (0.89) | 3.10 (1.94) | 3.49 (1.12) | 3.79 (0.82) | 3.57 (0.96) | 3.30 (1.70) |
| Show-o-Turbo | 3.48 (1.12) | 3.77 (0.88) | 3.53 (0.94) | 3.33 (1.60) | 3.50 (1.06) | 3.89 (0.86) | 3.64 (1.01) | 3.45 (1.44) |
| VARGPT | 2.60 (0.66) | 0.94 (2.37) | 0.94 (2.32) | 0.55 (1.99) | 2.55 (**0.30**) | 0.94 (2.39) | 0.87 (2.17) | 0.67 (2.16) |
| VARGPT-v1.1 | 3.13(0.83) | 0.89(1.98) | 1.16(2.11) | 0.72(1.83) | 3.26(0.76) | 1.01(2.12) | 1.23(1.95) | 0.68(1.96) |
| BAGEL | 2.97(0.83) | 3.92(0.81) | 4.18(0.75) | 2.81(1.21) | 3.11(0.91) | 3.89(0.72) | 4.23(0.66) | 2.87(1.33) |
| Gemini+Flux | 3.94 (0.94) | 4.06 (0.58) | 4.43 (**0.47**) | 3.81 (0.90) | 3.97 (0.57) | 4.12 (0.71) | **4.48** (0.64) | 3.84 (1.11) |
| GPT-4o+DALL-E | 4.05 (**0.37**) | 4.08 (**0.48**) | 4.41 (0.57) | 3.94 (0.64) | 3.99 (0.65) | 4.10 (0.81) | 4.45 (0.58) | 3.87 (1.16) |
| SEIR | **4.41** (0.55) | **4.46** (0.55) | **4.47** (0.53) | **4.51** (0.57) | **4.39** (0.61) | **4.49** (0.63) | 4.44 (**0.45**) | **4.53** (**0.51**) |

### 4.4 RELIABILITY OF SYNJUDGE

#### 4.4.1 EVALUATION SETUP FOR JUDGES

To identify a reliable automatic evaluator that aligns closely with human scoring, we conduct a comparative experiment. The protocol is designed to rigorously measure each candidate judge's deviation from a human-annotated ground truth.

We evaluate a total of five model-based judges against our human ground truth. The candidates are: (1) **Human Judge (Ground Truth):** A panel of ten experts whose scores serve as the gold standard. A cross-review protocol was used to ensure scoring reliability and mitigate individual bias. (2) **Zero-Shot MLLM Judges:** Three off-the-shelf MLLMs used for automated assessment: GPT-4o (OpenAI, 2024), QwenVL2.5 (Bai et al., 2025), and InternVL2.5 (Chen et al., 2024c). (3) **SynJudge Candidates (Finetuned):** We fine-tuned two strong MLLM backbone candidates, **QwenVL-trained** and **InternVL-trained**, to create our proposed evaluator.

The evaluation is conducted on a test set of 9,600 human-annotated question-answer pairs. To provide a comprehensive assessment of judge performance, we use two complementary metrics: (1) Root Mean Squared Error (RMSE): Measures the magnitude of the deviation from human scores. (2) Human Agreement (A@1): Measures the percentage of scores that are within a 1-point tolerance of the human rating, reflecting practical reliability. A lower RMSE and a higher A@1 indicate stronger alignment with human judgment. The SynJudge candidates were trained on a separate training set of 38,400 human-annotated pairs. Detailed definitions of the metrics and training hyperparameters of SynJudge are provided in the appendix (§D.2, §D.4).

### 4.4.2 COMPARISON RESULTS

Figure 4 and Table 6 report the performance of each judge using both RMSE and A@1 metrics. The results show a clear trend: finetuned judges decisively outperform zero-shot models on both metrics.

The **QwenVL-trained** judge demonstrates the strongest alignment with human preferences, achieving both the lowest average RMSE and the highest average A@1 of **95.4%**. This high agreement rate signifies that over 95% of its scores are within one point of human judgment, confirming its high reliability. In contrast, zero-shot judges like GPT-4o lag significantly, with A@1 scores around 86.5%. Based on this superior performance across complementary metrics, we select QwenVL-trained as our final **SynJudge**.

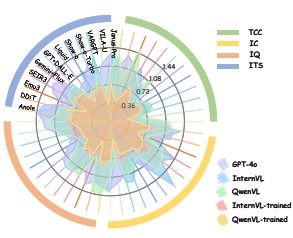

Figure 4: Visualization of RMSE for different judges.

Table 6: Judge performance comparison. We report average RMSE (lower is better) and Human Agreement (A@1, higher is better) against human scores. The best result in each row is highlighted in bold. QwenVL_trained demonstrates the strongest alignment.

| Dimension | GPT-4o | QwenVL | InternVL | QwenVL_trained | InternVL_trained |
|---|---|---|---|---|---|
| TCC (RMSE) | 1.01 | 0.81 | 0.96 | **0.54** | 0.55 |
| ICC (RMSE) | 1.02 | 1.09 | 0.90 | 0.72 | **0.70** |
| IQ (RMSE) | 0.98 | 0.96 | 1.06 | **0.68** | 0.72 |
| ITS (RMSE) | 1.18 | 1.20 | 1.03 | **0.67** | 0.72 |
| A@1 | 0.865 | 0.875 | 0.866 | **0.954** | 0.945 |

## 5 CONCLUSION

In this paper, we present InterSyn, a large scale, high quality multimodal dataset designed for instruction-following and interleaved image-text generation. Constructed via the fully automated SEIR method, InterSyn combines scale, diversity, and fidelity, supporting multi-turn dialogues where each response is refined to achieve not only semantic completeness but also tight image-text synergy—ensuring that visual and textual modalities complement each other to convey meaning collaboratively. To complement InterSyn, we introduce SynJudge, a multi-dimensional automatic evaluator specifically designed to assess interleaved outputs across four key dimensions, including a dedicated metric for image-text synergy. Unlike traditional metrics focused on surface-level alignment or consistency, SynJudge emphasizes the semantic interplay between images and text, rewarding complementary relationships while penalizing redundancy or disjointness. Extensive experiments validate the effectiveness of both InterSyn and SEIR. Models fine-tuned on InterSyn consistently outperform strong baselines, showing notable improvements in instruction alignment, multimodal reasoning, and especially the ability to produce coherent, synergistic interleaved content. We believe this work lays a solid foundation for future research in scalable multimodal data generation, robust synergy-centric evaluation, and the development of general-purpose multimodal intelligence systems that understand and communicate across modalities in a truly integrated manner.

**Reproducibility Statement.** To facilitate the verification of our findings and to support future research, we are committed to making our work fully reproducible. The complete codebase, including scripts for the SEIR data generation pipeline, SynJudge training, and all evaluation protocols, will be made publicly available on GitHub upon publication. We will also release the full InterSyn dataset (1.8M samples), all benchmark sets used for our main experiments, and the trained weights of our final SynJudge evaluator. The core of our methodology relies on publicly available models, and all prompts, model configurations, and hyperparameters are extensively detailed in the appendix to ensure that our results can be precisely replicated.

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

## A STATEMENT ON LLMs USAGE

The authors used large language models (LLMs) during the writing process solely for language refinement and editing. It should be explicitly stated that LLMs were not employed in any core aspects of the research, including the formulation of research ideas, the design of methodologies, the execution of experiments, or the development of conclusions. All scholarly contributions were made independently by the authors.

## B APPENDIX OVERVIEW

The all supplementary document is organized as follows:

- Comparison of datasets and Samples of InterSyn are shown in §C.
- Supplementary analysis of experimental Data are shown in §D.
- Question templates and topic hierarchy are shown in §E.
- The evaluation dimensions for question and answer are shown in §F.
- Prompts used in this work are shown in §G.
- The benchmark samples are shown in §H.
- Human annotation platform are shown in §I.
- Limitations of this study are shown in §J.

## C COMPARISON OF DATASETS AND INTERSYN SAMPLES

### C.1 COMPARISON OF DATASETS

The table provides a comprehensive comparison of InterSyn with representative multimodal datasets and benchmarks. Existing datasets such as MMC4 and OBELICS primarily rely on large scale web-crawled corpora, often lacking instruction-following capabilities and multi-turn structures. Other resources like CoMM and ShareGPT4V improve data cleanliness but remain limited to single-turn interactions without tight semantic supervision.

Recent efforts including LeafInstruct introduce instruction-following supervision but still operate in single-turn formats. Meanwhile, benchmark-oriented resources—such as OpenLEAF, ISG-BENCH, MMIE, InterleavedBench, and OpenING—focus on evaluating generation quality but are constrained by small scale and limited turn complexity.

In contrast, InterSyn is the first to offer a large scale, multi-turn, instruction-following dataset specifically designed for interleaved image-text generation. Built with the SEIR method, InterSyn not only ensures high quality visual-textual synergy but also scales to 1.8 million samples—orders of magnitude larger than existing benchmarks. Its emphasis on dialogue coherence, iterative refinement, and synergistic multimodal responses fills a critical gap in current resources, laying the groundwork for developing and evaluating truly unified multimodal generation models.

Table 7: Comparison of Multimodal Datasets and Benchmarks. Abbreviations: Cat. = Category; Inst. = *instruction-following*; MT. = *multi-turn*; DS. = *dataset*; BM. = *benchmark*; Gen. = *generation*.

| Name | Cat. | Source | Method | Size | Inst. | MT. |
|---|---|---|---|---|---|---|
| **InterSyn** | DS. | Collected questions | SEIR | **1.8M** | ✓ | ✓ |
| MMC4 | DS. | Common Crawl | CLIP-based filtering | 101.2M Doc. | ✗ | ✗ |

| Name | Cat. | Source | Method | Size | Inst. | MT. |
|------|------|--------|--------|------|-------|-----|
| OBELICS | DS. | Common Crawl | Multi-granularity filtering | 141M pages | ✗ | ✗ |
| CoMM | DS. | WikiHow, StoryBird, eHow, etc. | Multi-perspective filtering | 227K Doc. | ✗ | ✗ |
| ShareGPT4V | DS. | GPT-4V captions | Share-Captioner | 1.2M pairs | ✗ | ✗ |
| LeafInstruct | DS. | MMC4, VIST, YouCook2, etc. | Text & image quality filtering | 38,272 | ✓ | ✗ |
| OpenLEAF | BM. | User Queries | GPT-4 Gen. and human review | 660 | ✓ | ✗ |
| ISG-BENCH | BM. | VIST, CoMM, manual Gen. | Model Gen. & human review | 1,150 | ✓ | ✗ |
| MMIE | BM. | WikiHow, VIST, MathVista, etc. | Sampling & reconstruction | 20,103 | ✓ | ✗ |
| InterleavedBench | BM. | VIST, WikiHow, etc. | GPT-4o Gen. + human review | 815 | ✓ | ✗ |
| OpenING | BM. | YouTube, Google, etc. | Manual pipeline | 5,400 | ✓ | ✗ |

## C.2 SINGLE-TURN SAMPLES

Samples of data are shown in Figure 5

## C.3 MULTI-TURN SAMPLES

Samples of data are shown in Figure 6

# D SUPPLEMENTARY ANALYSIS OF EXPERIMENTAL

## D.1 SYMBOLS AND NOTATIONS FOR SEIR METHOD

We summarize the key symbols used in the SEIR method below:

- $\mathcal{T}$: Set of question templates.
- $\mathcal{Z}$: Set of topics.
- $T \in \mathbb{N}^+$: Number of conversation turns.
- $K \in \mathbb{N}^+$: Number of refinement iterations at each stage.
- $\mathcal{H}^{(t-1)}$: History of the conversation up to turn $t-1$, represented as $\{(q^{(i)}, a^{(i)}, i^{(i)})\}_{i=1}^{t-1}$.
- $q^{(t)}$: **Final** question generated at conversation turn $t$.
- $q_k^{(t)}$: Question after $k$ refinement iterations at conversation turn $t$.
- $a^{(t)}$: **Final** text answer generated at conversation turn $t$.
- $a_k^{(t)}$: Text answer after $k$ refinement iterations at conversation turn $t$.
- $\gamma^{(t)}$: Temporary caption associated with the text answer at conversation turn $t$.
- $\gamma_k^{(t)}$: Temporary caption after $k$ refinement iterations.

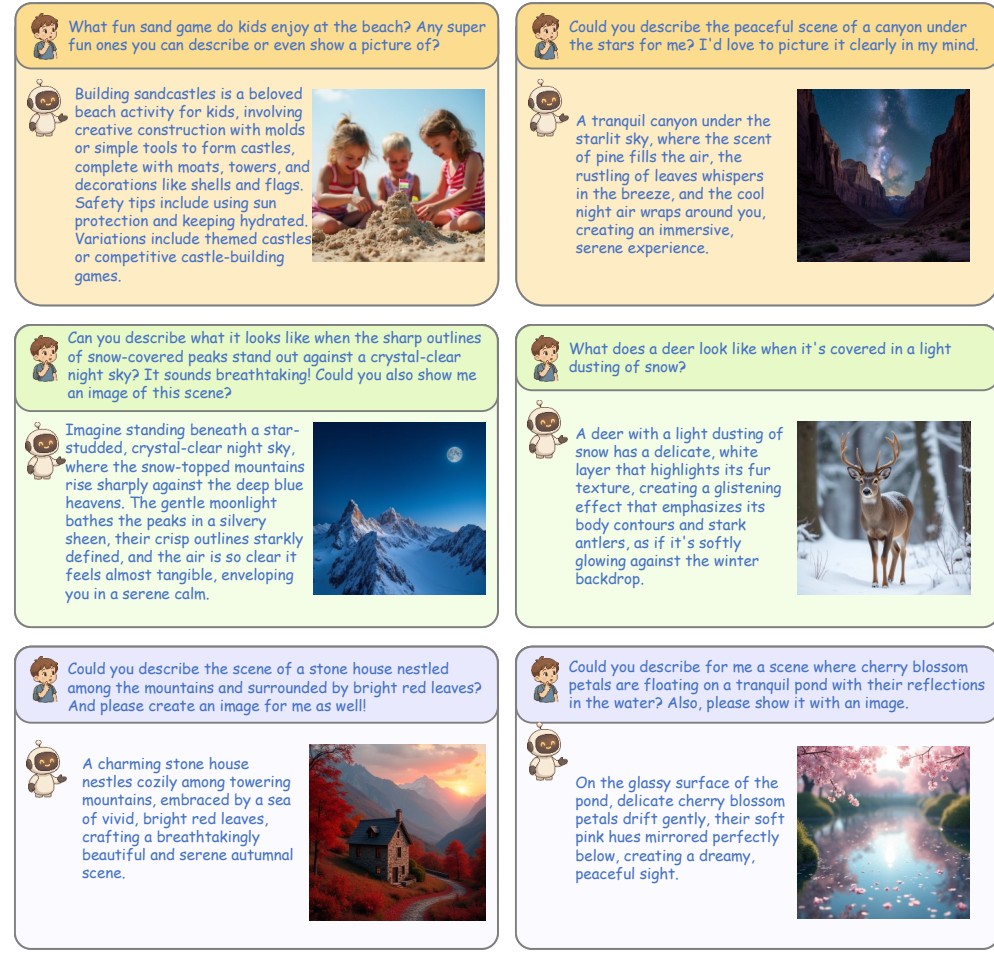

Figure 5: Examples of single-turn conversation

- $c^{(t)}$: **Final** image caption at conversation turn $t$.

- $c_k^{(t)}$: Image caption after $k$ refinement iterations.

- $I^{(t)}$: **Final** generated image at conversation turn $t$.

- $I_k^{(t)}$: Generated image after $k$ refinement iterations.

- $\mathcal{M}_L$: Language model used for text generation and refinement.

- $\mathcal{M}_V$: Vision-language model used for image caption evaluation and refinement.

- $\mathcal{M}_G$: Text-to-image generation model.

- $p_g(\cdot)$: Prompt function for generating model response.

- $p_s(\cdot)$: Prompt function for generating refinement suggestions.

- $p_r(\cdot)$: Prompt function for applying refinements.

## D.2 EVALUATION METRICS FOR JUDGES AND GENERATORS

To facilitate a quantitative evaluation of judges and generators, we design a set of metrics.

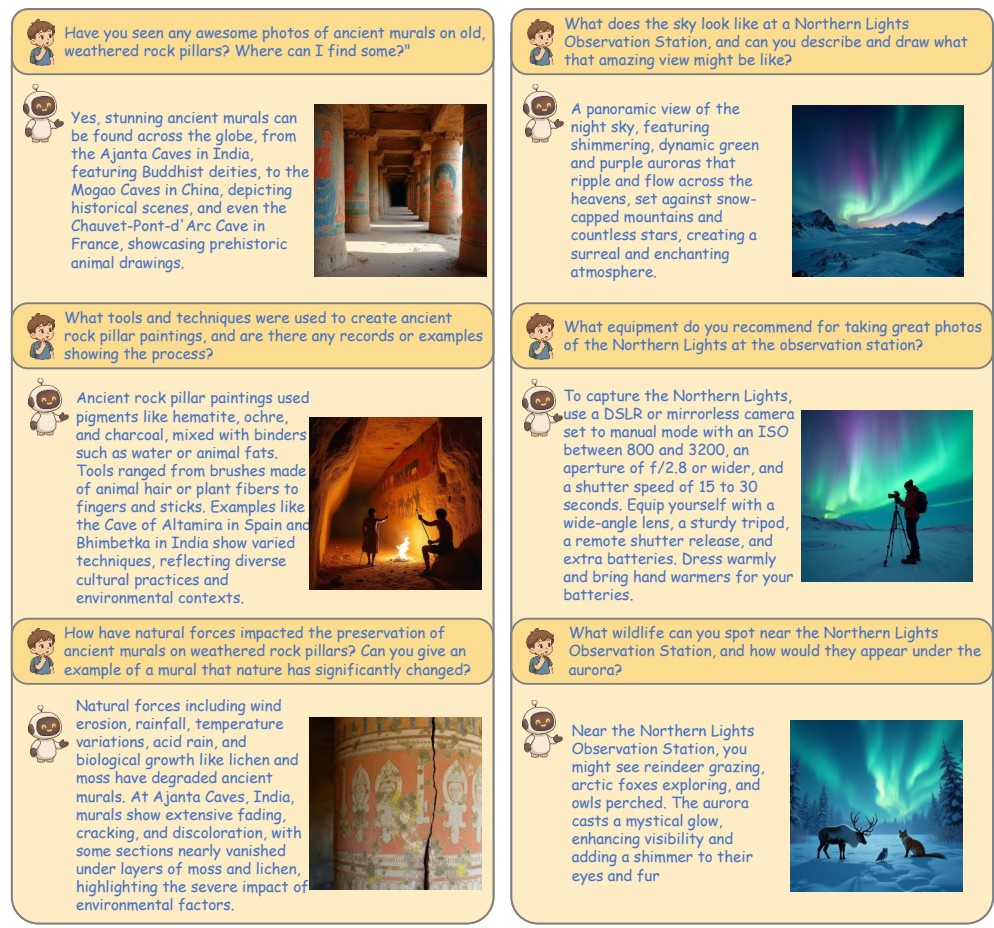

Figure 6: Examples of multi-turn conversation

**Mean Calculation**    Let $x_{i,d}$ denote the score given by a judge to the $i$-th sample generated by a generator, under evaluation dimension $d$, and let $N$ be the total number of samples. Then, for each (judge, generator) pair, the mean score is computed as:

$$S_d = \frac{1}{N} \sum_{i=1}^{N} x_{i,d} \tag{5}$$

The mean score $S_d$ reflects the average performance of a generator, as evaluated by a specific judge under dimension $d$.

**Variance Calculation**    To estimate the variability of the scores, we compute the variance:

$$\sigma_d = \frac{1}{N} \sum_{i=1}^{N} (x_{i,d} - S_d)^2 \tag{6}$$

The variance $\sigma_d$ captures the consistency of the generator's performance across different questions. A higher variance indicates greater inconsistency in quality.

**Root Mean Squared Error (RMSE)** To measure the agreement between a model-based judge $M$ and human judge $H$, we compute the RMSE between their respective scores for each sample:

$$\text{RMSE}_d = \sqrt{\frac{1}{N} \sum_{i=1}^{N} \left( x_{i,d}^M - x_{i,d}^H \right)^2} \qquad (7)$$

Here, $x_{i,d}^M$ and $x_{i,d}^H$ denote the scores assigned by the model-based and human judge respectively. RMSE quantifies the deviation between a model-based judge's scores and those of human judge in dimension $d$. Lower RMSE values indicate higher alignment with human preferences, and thus higher reliability of the model-based judge.

**Human Agreement within Tolerance (A@$\tau$)** While RMSE measures the magnitude of error, it can be sensitive to outliers. To provide a complementary view of judge reliability, we also introduce the Human Agreement within Tolerance (A@$\tau$) metric. This metric calculates the percentage of evaluations where the judge's score falls within a specified tolerance margin, $\tau$, of the human score. Given the subjective nature of the scoring task, we set a tolerance of $\tau = 1$ point. The metric, A@1, is calculated as:

$$\text{A@1} = \frac{1}{N} \sum_{i=1}^{N} \mathbb{I}(|x_{i,d}^M - x_{i,d}^H| \leq 1) \qquad (8)$$

where $\mathbb{I}(\cdot)$ is the indicator function, which is 1 if the condition is true and 0 otherwise. A higher A@1 score indicates that the judge's scores are more frequently in close agreement with human evaluators, reflecting greater practical reliability.

This evaluation framework provides a comprehensive analysis of the performance of all the generators across multiple dimensions, ensuring objective comparison from both human and model-based perspectives.

### D.3 Models Used in the SEIR Method

To construct the dataset, we adopt Qwen2.5-32B-Instruct as the language model ($\mathcal{M}_L$), InternVL2_5-26B as the vision-language model ($\mathcal{M}_V$), and FLUX.1-dev as the text-to-image generation model ($\mathcal{M}_G$). These open-source models are used as the default configuration for our SEIR framework. Importantly, the framework is modular by design—each component can be substituted with other models, offering flexibility for different deployment environments or research needs.

To assess the generality and robustness of SEIR, we systematically evaluated its performance across a range of model configurations. Specifically, we experimented with different combinations of open-source LMs (InternLM, DeepSeek-R1, Qwen) and VLMs (InternVL, QwenVL), while keeping the generative model fixed as Flux. In addition, to benchmark against high-performing closed-source alternatives, we included a configuration that uses GPT-4o as both the LM and VLM, and DALL-E3 as the generative model.

The results, presented in Table 8, demonstrate several key trends:

- **SEIR consistently improves data quality.** Across all configurations, we observe noticeable improvements in all four evaluation dimensions—TCC, ICC, IQ and ITS—after applying SEIR. The gains are particularly significant in the TCC, ICC and ITS dimensions, reflecting SEIR's ability to enhance the semantic alignment and cooperative informativeness of multimodal outputs.

- **Initial quality varies across model combinations.** Among the open-source configurations, those involving Qwen (e.g., Qwen+InternVL or QwenVL) generally exhibit stronger performance in the No_SEIR stage. In contrast, InternLM+QwenVL and DeepSeek-R1 + InternVL show relatively weaker initial consistency, suggesting differences in language-vision alignment quality across model families.

- **SEIR is especially effective for lower-performing combinations.** The relative improvements are more pronounced for model combinations with lower baseline performance. For example, InternLM + QwenVL improves by 0.60 in TCC and 0.70 in ITS, and DeepSeek-R1+InternVL shows notable gains across all metrics despite its initially modest performance.

- **SEIR narrows the gap between open- and closed-source models.** While the GPT-4o+DALL-E3 configuration achieves the highest initial quality across all metrics, the application of SEIR allows open-source configurations to reach comparable performance levels. For instance, Qwen+InternVL + Flux achieves 4.42 (TCC) and 4.51 (ITS) after SEIR, which closely rivals the 4.44 and 4.54 obtained by GPT-4o+DALL-E3.

- **Text-Image Synergy (ITS) shows the largest variance.** This dimension benefits the most from SEIR optimization, particularly in cases where image-text redundancy or disconnection was prevalent before refinement. The improvement indicates SEIR's effectiveness in jointly adjusting both modalities to produce more complementary multimodal answers.

Overall, these results confirm that SEIR is a robust and generalizable enhancement framework. It consistently improves dataset quality across a wide range of model backbones, and significantly reduces the reliance on expensive closed-source models. Consequently, we adopt the open-source setup of Qwen+InternVL+Flux as our default configuration, balancing quality, flexibility, and cost-effectiveness. During the dataset generation and experimental testing process, this work consumed approximately 90,000 H100 hours.

Table 8: Comparison of dataset quality before and after SEIR optimization using different model configurations.

| method | LM | VLM | GM | TCC | ICC | IQ | ITS |
|---|---|---|---|---|---|---|---|
| No_SEIR | InternLM | InternVL | Flux | 3.72 | 3.98 | 4.30 | 3.70 |
| SEIR | InternLM | InternVL | Flux | 4.34 | 4.35 | 4.38 | 4.42 |
| No_SEIR | InternLM | QwenVL | Flux | 3.66 | 3.89 | 4.28 | 3.68 |
| SEIR | InternLM | QwenVL | Flux | 4.26 | 4.32 | 4.35 | 4.38 |
| No_SEIR | DeepSeek-R1 | InternVL | Flux | 3.65 | 3.82 | 4.37 | 3.66 |
| SEIR | DeepSeek-R1 | InternVL | Flux | 4.20 | 4.28 | 4.43 | 4.19 |
| No_SEIR | Qwen | QwenVL | Flux | 3.80 | 3.93 | 4.35 | 3.75 |
| SEIR | Qwen | QwenVL | Flux | 4.37 | 4.40 | 4.36 | 4.52 |
| No_SEIR | Qwen | InternVL | Flux | 3.85 | 4.01 | 4.42 | 3.79 |
| SEIR | Qwen | InternVL | Flux | 4.42 | 4.47 | 4.44 | 4.51 |
| No_SEIR | GPT-4o | GPT-4o | DALL-E3 | 4.05 | 4.08 | 4.41 | 3.94 |
| SEIR | GPT-4o | GPT-4o | DALL-E3 | 4.44 | 4.46 | 4.43 | 4.54 |

## D.4 Hyperparameters Used for Training Judge Model

We fine-tuned two large multimodal models, InternVL2.5-8B and QwenVL2.5-7B, as judge models to evaluate the interleaved image-text content completeness. We followed common practices for large scale model fine-tuning, applying weight decay regularization, learning rate warmup, and gradient clipping to ensure training stability. All experiments were conducted using mixed-precision training on distributed GPU clusters.

For the QwenVL2.5-7B judge model, we adopted a multi-GPU training setup using 4 devices with a total training batch size of 8, obtained by setting a per-device batch size of 1 and a gradient accumulation step of 2. The model was trained using the AdamW optimizer with $\beta_1 = 0.9$, $\beta_2 = 0.999$, and $\epsilon = 1 \times 10^{-8}$. The initial learning rate was set to $1 \times 10^{-5}$ and scheduled using a cosine decay strategy with a warmup ratio of 10%. A fixed random seed of 42 was used for reproducibility. Evaluation was conducted using a batch size of 8 per device, resulting in a total evaluation batch size of 32.

For the InternVL2.5-8B referee model, we adopted a multi-GPU training setup, using 4 devices with a total training batch size of 8. This was achieved by setting the batch size per device to 1 and the gradient accumulation steps to 2. The model was trained using the AdamW optimizer, with

$\beta_1 = 0.9$, $\beta_2 = 0.999$, $\epsilon = 1 \times 10^{-8}$, a gradient clipping threshold of 1.0, and a weight decay of 0.05. The initial learning rate was set to $4 \times 10^{-5}$, and a cosine annealing strategy was employed for adjustment, with a warm-up ratio of 3%. For reproducibility, a fixed random seed of 42 was used. During evaluation, the batch size per device was 8, resulting in a total evaluation batch size of 32.

### D.5   FURTHER VERIFICATION OF SEIR EFFECTIVENESS

To investigate the impact of the SEIR method, we conduct a fine-grained ablation by varying AR and IR iterations. Results in Table 9 show that both AR and IR contribute positively to performance. Specifically, increasing AR iterations mainly improves TCC, ICC, and ITS, while IR iterations further enhance ICC and ITS. These results confirm that AR enriches textual content and coherence, while IR reinforces visual relevance and multimodal synergy.

Table 9: Ablation study: impact of AR and IR iterations on Anole and VILA-U. "null" indicates baseline performance without SEIR-based training.

(a) Performance of Anole.

| AR | IR | TCC | ICC | IQ | ITS |
|----|----|-----|-----|-----|-----|
| null | null | 3.09 | 3.01 | 2.92 | 2.26 |
| 0 | 0 | 3.33 | 3.17 | 2.92 | 2.77 |
| 1 | 1 | 3.37 | 3.21 | 3.07 | 2.71 |
| 2 | 2 | 3.41 | 3.27 | 3.03 | 2.79 |
| 3 | 0 | 3.47 | 3.20 | 3.03 | 2.85 |
| 3 | 1 | 3.51 | 3.25 | 3.11 | 2.93 |
| 3 | 2 | 3.49 | 3.3 | 3.08 | 2.91 |
| 3 | 3 | 3.52 | 3.32 | 3.1 | 2.94 |

(b) Performance of VILA-U.

| AR | IR | TCC | ICC | IQ | ITS |
|----|----|-----|-----|-----|-----|
| null | null | 2.46 | 3.72 | 3.37 | 2.19 |
| 0 | 0 | 3.07 | 3.83 | 3.37 | 3.13 |
| 1 | 0 | 3.1 | 3.8 | 3.34 | 3.2 |
| 2 | 0 | 3.17 | 3.79 | 3.39 | 3.27 |
| 3 | 0 | 3.17 | 3.81 | 3.36 | 3.27 |
| 3 | 1 | 3.17 | 3.81 | 3.4 | 3.3 |
| 3 | 2 | 3.21 | 3.85 | 3.37 | 3.29 |
| 3 | 3 | 3.19 | 3.83 | 3.39 | 3.33 |

### D.6   SYNJUDGE TRAIN/TEST DATA

Our full annotated data contains 48,000 samples. These 48,000 questions were generated by the SEIR method through three iterations, with the question template and topic hierarchy ensuring sufficient diversity across topics and conversational query style.

To obtain a broad distribution of multimodal question–answer (QA) outputs, these 48,000 questions were randomly assigned to different generators, which then produced responses. This strategy ensures that the resulting QA pairs cover a wide quality spectrum. After generation, all question–answer pairs were annotated by trained human annotators for TCC, ICC, IQ, and ITS over a two-week period.

The annotated dataset was split into a training set (80%) and a test set (20%). The training set was used to fine-tune candidate judge models (e.g., QwenVL_trained, InternVL_trained), while the test set was reserved for evaluation. For the evaluation, we used RMSE as the metric to measure how closely the scores from different judges align with ground-truth human annotations: a lower RMSE indicates a better judge. This evaluation process is what leads to the selection of our final model, SynJudge.

### D.7   DETAILED ANALYSIS OF THE JUDGE'S SCORING

**Judge Deviation Analysis.**   To better understand each judge's scoring behavior, we report the distribution of absolute score differences in Tables 10–14. Each table shows, for a given judge, the proportion of samples where the model's score differs from the human reference by 0 to 5. These detailed distributions provide a fine-grained view of the judges' alignment with human evaluators.

**Judge Agreement Analysis.**   Table 15 provides a detailed breakdown of the **Human Agreement within Tolerance (A@1)** for each judge across all evaluated generators and dimensions. As formally defined in Appendix §D.2, this metric reflects the proportion of scores where the absolute difference between the model and human judge is no more than one point ($\tau = 1$), which we consider an acceptable margin for subjective tasks.

The results reinforce our findings from the main paper. The finetuned judges consistently achieve higher agreement rates. QwenVL_trained achieves the highest average A@1 of 95.4%. InternVL_trained also shows strong performance at 94.5%, significantly better than the non-finetuned baselines. In contrast, zero-shot models like GPT-4o and QwenVL exhibit noticeably lower agreement, around 86.5% and 87.5% respectively, indicating that they are less reliable for fine-grained evaluation without specialized tuning. These findings further justify our selection of QwenVL_trained as the backbone for **SynJudge**.

Table 10: Gap proportion between GPT-4o and human scoring.

| Model | Dimension | Score | | | | | |
|---|---|---|---|---|---|---|---|
| | | 0 | 1 | 2 | 3 | 4 | 5 |
| Anole | TCC | 0.463 | 0.343 | 0.159 | 0.031 | 0.004 | 0.0 |
| | ICC | 0.525 | 0.325 | 0.106 | 0.037 | 0.007 | 0.0 |
| | IQ | 0.52 | 0.309 | 0.128 | 0.036 | 0.007 | 0.0 |
| | ITS | 0.536 | 0.289 | 0.111 | 0.056 | 0.007 | 0.001 |
| GPT-4o+DALL-E | TCC | 0.654 | 0.318 | 0.025 | 0.003 | 0.0 | 0.0 |
| | ICC | 0.686 | 0.276 | 0.031 | 0.003 | 0.002 | 0.002 |
| | IQ | 0.636 | 0.295 | 0.065 | 0.002 | 0.0 | 0.002 |
| | ITS | 0.635 | 0.303 | 0.039 | 0.019 | 0.002 | 0.002 |
| DDiT | TCC | 0.938 | 0.024 | 0.017 | 0.012 | 0.009 | 0.0 |
| | ICC | 0.467 | 0.379 | 0.111 | 0.037 | 0.006 | 0.0 |
| | IQ | 0.434 | 0.404 | 0.135 | 0.023 | 0.004 | 0.0 |
| | ITS | 0.843 | 0.043 | 0.023 | 0.022 | 0.027 | 0.042 |
| Emu3 | TCC | 0.5 | 0.398 | 0.085 | 0.016 | 0.001 | 0.0 |
| | ICC | 0.535 | 0.333 | 0.082 | 0.037 | 0.012 | 0.001 |
| | IQ | 0.546 | 0.367 | 0.074 | 0.012 | 0.001 | 0.0 |
| | ITS | 0.454 | 0.358 | 0.123 | 0.052 | 0.013 | 0.0 |
| SEIR | TCC | 0.589 | 0.329 | 0.066 | 0.013 | 0.002 | 0.001 |
| | ICC | 0.686 | 0.251 | 0.037 | 0.017 | 0.009 | 0.0 |
| | IQ | 0.665 | 0.281 | 0.051 | 0.001 | 0.001 | 0.001 |
| | ITS | 0.577 | 0.3 | 0.063 | 0.037 | 0.016 | 0.007 |
| Gemini+Flux | TCC | 0.636 | 0.311 | 0.037 | 0.011 | 0.003 | 0.002 |
| | ICC | 0.705 | 0.237 | 0.036 | 0.015 | 0.007 | 0.0 |
| | IQ | 0.695 | 0.256 | 0.047 | 0.002 | 0.0 | 0.0 |
| | ITS | 0.657 | 0.267 | 0.045 | 0.023 | 0.006 | 0.002 |
| Janus-Pro | TCC | 0.31 | 0.419 | 0.234 | 0.036 | 0.001 | 0.0 |
| | ICC | 0.412 | 0.395 | 0.124 | 0.06 | 0.009 | 0.0 |
| | IQ | 0.428 | 0.369 | 0.164 | 0.034 | 0.005 | 0.0 |
| | ITS | 0.369 | 0.407 | 0.131 | 0.078 | 0.014 | 0.001 |
| Liquid | TCC | 0.465 | 0.389 | 0.12 | 0.024 | 0.002 | 0.0 |
| | ICC | 0.513 | 0.368 | 0.073 | 0.039 | 0.006 | 0.001 |
| | IQ | 0.478 | 0.399 | 0.094 | 0.025 | 0.002 | 0.002 |
| | ITS | 0.357 | 0.414 | 0.14 | 0.071 | 0.016 | 0.002 |
| Show-o | TCC | 0.525 | 0.343 | 0.104 | 0.026 | 0.002 | 0.0 |
| | ICC | 0.475 | 0.367 | 0.101 | 0.043 | 0.013 | 0.001 |
| | IQ | 0.445 | 0.382 | 0.139 | 0.029 | 0.005 | 0.0 |
| | ITS | 0.471 | 0.331 | 0.114 | 0.06 | 0.019 | 0.005 |
| Show-o-Turbo | TCC | 0.432 | 0.34 | 0.17 | 0.056 | 0.002 | 0.0 |
| | ICC | 0.53 | 0.332 | 0.095 | 0.032 | 0.011 | 0.0 |
| | IQ | 0.433 | 0.383 | 0.16 | 0.019 | 0.005 | 0.0 |
| | ITS | 0.457 | 0.36 | 0.122 | 0.044 | 0.013 | 0.004 |
| VARGPT | TCC | 0.351 | 0.356 | 0.235 | 0.051 | 0.007 | 0.0 |
| | ICC | 0.861 | 0.064 | 0.037 | 0.03 | 0.005 | 0.003 |
| | IQ | 0.875 | 0.082 | 0.034 | 0.003 | 0.003 | 0.003 |
| | ITS | 0.864 | 0.071 | 0.032 | 0.024 | 0.008 | 0.001 |
| VILA-U | TCC | 0.617 | 0.249 | 0.105 | 0.024 | 0.004 | 0.001 |
| | ICC | 0.428 | 0.405 | 0.117 | 0.046 | 0.004 | 0.0 |
| | IQ | 0.403 | 0.37 | 0.172 | 0.047 | 0.007 | 0.001 |
| | ITS | 0.562 | 0.254 | 0.1 | 0.038 | 0.026 | 0.02 |

Table 11: Gap proportion between InternVL and human scoring.

| Model | Dimension | Score | | | | | |
|---|---|---|---|---|---|---|---|
| | | 0 | 1 | 2 | 3 | 4 | 5 |
| Anole | TCC | 0.523 | 0.294 | 0.147 | 0.034 | 0.001 | 0.001 |
| | ICC | 0.574 | 0.286 | 0.106 | 0.03 | 0.003 | 0.001 |
| | IQ | 0.504 | 0.272 | 0.176 | 0.042 | 0.005 | 0.001 |
| | ITS | 0.592 | 0.26 | 0.116 | 0.022 | 0.01 | 0.0 |
| GPT-4o+DALL-E | TCC | 0.658 | 0.326 | 0.015 | 0.001 | 0.0 | 0.0 |
| | ICC | 0.709 | 0.248 | 0.037 | 0.003 | 0.001 | 0.002 |
| | IQ | 0.647 | 0.269 | 0.075 | 0.006 | 0.0 | 0.003 |
| | ITS | 0.653 | 0.281 | 0.041 | 0.021 | 0.0 | 0.004 |
| DDiT | TCC | 0.942 | 0.022 | 0.02 | 0.005 | 0.01 | 0.001 |
| | ICC | 0.574 | 0.327 | 0.081 | 0.015 | 0.003 | 0.0 |
| | IQ | 0.485 | 0.347 | 0.136 | 0.029 | 0.003 | 0.0 |
| | ITS | 0.927 | 0.025 | 0.026 | 0.01 | 0.01 | 0.002 |
| Emu3 | TCC | 0.586 | 0.329 | 0.075 | 0.008 | 0.002 | 0.0 |
| | ICC | 0.629 | 0.27 | 0.072 | 0.022 | 0.006 | 0.001 |
| | IQ | 0.531 | 0.328 | 0.12 | 0.021 | 0.0 | 0.0 |
| | ITS | 0.496 | 0.351 | 0.105 | 0.044 | 0.004 | 0.0 |
| SEIR | TCC | 0.589 | 0.352 | 0.048 | 0.01 | 0.001 | 0.0 |
| | ICC | 0.708 | 0.239 | 0.039 | 0.012 | 0.001 | 0.001 |
| | IQ | 0.692 | 0.246 | 0.057 | 0.003 | 0.001 | 0.001 |
| | ITS | 0.611 | 0.271 | 0.075 | 0.031 | 0.009 | 0.003 |
| Gemini+Flux | TCC | 0.629 | 0.332 | 0.032 | 0.004 | 0.002 | 0.001 |
| | ICC | 0.705 | 0.241 | 0.045 | 0.005 | 0.004 | 0.0 |
| | IQ | 0.694 | 0.248 | 0.056 | 0.002 | 0.0 | 0.0 |
| | ITS | 0.633 | 0.279 | 0.066 | 0.018 | 0.002 | 0.002 |
| Janus-Pro | TCC | 0.37 | 0.348 | 0.235 | 0.046 | 0.001 | 0.0 |
| | ICC | 0.52 | 0.327 | 0.124 | 0.026 | 0.003 | 0.0 |
| | IQ | 0.378 | 0.31 | 0.238 | 0.066 | 0.008 | 0.0 |
| | ITS | 0.471 | 0.313 | 0.161 | 0.049 | 0.006 | 0.0 |
| Liquid | TCC | 0.438 | 0.389 | 0.139 | 0.03 | 0.003 | 0.001 |
| | ICC | 0.552 | 0.339 | 0.076 | 0.03 | 0.003 | 0.0 |
| | IQ | 0.42 | 0.401 | 0.137 | 0.033 | 0.006 | 0.003 |
| | ITS | 0.433 | 0.343 | 0.149 | 0.059 | 0.013 | 0.003 |
| Show-o | TCC | 0.592 | 0.314 | 0.074 | 0.02 | 0.0 | 0.0 |
| | ICC | 0.556 | 0.319 | 0.1 | 0.02 | 0.004 | 0.001 |
| | IQ | 0.473 | 0.308 | 0.181 | 0.032 | 0.006 | 0.0 |
| | ITS | 0.49 | 0.319 | 0.129 | 0.05 | 0.008 | 0.004 |
| Show-o-Turbo | TCC | 0.528 | 0.333 | 0.118 | 0.02 | 0.001 | 0.0 |
| | ICC | 0.581 | 0.316 | 0.082 | 0.017 | 0.004 | 0.0 |
| | IQ | 0.447 | 0.324 | 0.199 | 0.023 | 0.006 | 0.001 |
| | ITS | 0.492 | 0.347 | 0.113 | 0.038 | 0.007 | 0.003 |
| VARGPT | TCC | 0.271 | 0.368 | 0.287 | 0.062 | 0.009 | 0.003 |
| | ICC | 0.857 | 0.069 | 0.057 | 0.011 | 0.003 | 0.003 |
| | IQ | 0.857 | 0.083 | 0.043 | 0.011 | 0.003 | 0.003 |
| | ITS | 0.872 | 0.051 | 0.055 | 0.014 | 0.007 | 0.001 |
| VILA-U | TCC | 0.675 | 0.238 | 0.069 | 0.015 | 0.002 | 0.001 |
| | ICC | 0.553 | 0.316 | 0.099 | 0.028 | 0.004 | 0.0 |
| | IQ | 0.408 | 0.303 | 0.205 | 0.076 | 0.005 | 0.003 |
| | ITS | 0.684 | 0.195 | 0.087 | 0.023 | 0.009 | 0.002 |

Table 12: Gap proportion between QwenVL and human scoring.

| Model | Dimension | Score | | | | | |
|---|---|---|---|---|---|---|---|
| | | 0 | 1 | 2 | 3 | 4 | 5 |
| Anole | TCC | 0.783 | 0.189 | 0.027 | 0.001 | 0.0 | 0.0 |
| | ICC | 0.558 | 0.258 | 0.092 | 0.06 | 0.022 | 0.01 |
| | IQ | 0.577 | 0.294 | 0.102 | 0.022 | 0.005 | 0.0 |
| | ITS | 0.592 | 0.189 | 0.121 | 0.061 | 0.032 | 0.005 |
| GPT-4o+DALL-E | TCC | 0.679 | 0.299 | 0.022 | 0.0 | 0.0 | 0.0 |
| | ICC | 0.716 | 0.234 | 0.041 | 0.006 | 0.001 | 0.002 |
| | IQ | 0.579 | 0.325 | 0.081 | 0.012 | 0.003 | 0.0 |
| | ITS | 0.571 | 0.296 | 0.094 | 0.029 | 0.004 | 0.006 |
| DDiT | TCC | 0.947 | 0.021 | 0.013 | 0.008 | 0.01 | 0.001 |
| | ICC | 0.512 | 0.339 | 0.107 | 0.035 | 0.007 | 0.0 |
| | IQ | 0.484 | 0.317 | 0.17 | 0.029 | 0.0 | 0.0 |
| | ITS | 0.939 | 0.021 | 0.017 | 0.008 | 0.013 | 0.002 |
| Emu3 | TCC | 0.628 | 0.316 | 0.05 | 0.006 | 0.0 | 0.0 |
| | ICC | 0.561 | 0.312 | 0.039 | 0.041 | 0.031 | 0.016 |
| | IQ | 0.58 | 0.345 | 0.073 | 0.002 | 0.0 | 0.0 |
| | ITS | 0.543 | 0.277 | 0.118 | 0.043 | 0.015 | 0.004 |
| SEIR | TCC | 0.574 | 0.339 | 0.076 | 0.009 | 0.002 | 0.0 |
| | ICC | 0.67 | 0.256 | 0.051 | 0.013 | 0.007 | 0.003 |
| | IQ | 0.667 | 0.264 | 0.063 | 0.003 | 0.001 | 0.002 |
| | ITS | 0.566 | 0.283 | 0.082 | 0.042 | 0.017 | 0.01 |
| Gemini+Flux | TCC | 0.607 | 0.354 | 0.031 | 0.004 | 0.003 | 0.001 |
| | ICC | 0.548 | 0.358 | 0.04 | 0.027 | 0.011 | 0.016 |
| | IQ | 0.677 | 0.254 | 0.064 | 0.003 | 0.002 | 0.0 |
| | ITS | 0.613 | 0.325 | 0.042 | 0.012 | 0.004 | 0.004 |
| Janus-Pro | TCC | 0.433 | 0.343 | 0.196 | 0.027 | 0.001 | 0.0 |
| | ICC | 0.454 | 0.29 | 0.166 | 0.054 | 0.025 | 0.011 |
| | IQ | 0.429 | 0.35 | 0.161 | 0.053 | 0.006 | 0.001 |
| | ITS | 0.502 | 0.276 | 0.132 | 0.059 | 0.028 | 0.003 |
| Liquid | TCC | 0.621 | 0.298 | 0.064 | 0.015 | 0.002 | 0.0 |
| | ICC | 0.571 | 0.311 | 0.089 | 0.024 | 0.004 | 0.001 |
| | IQ | 0.656 | 0.283 | 0.044 | 0.013 | 0.004 | 0.0 |
| | ITS | 0.478 | 0.239 | 0.115 | 0.067 | 0.09 | 0.011 |
| Show-o | TCC | 0.542 | 0.35 | 0.088 | 0.016 | 0.004 | 0.0 |
| | ICC | 0.536 | 0.316 | 0.105 | 0.036 | 0.004 | 0.003 |
| | IQ | 0.457 | 0.364 | 0.143 | 0.029 | 0.006 | 0.001 |
| | ITS | 0.571 | 0.26 | 0.094 | 0.051 | 0.021 | 0.003 |
| Show-o-Turbo | TCC | 0.436 | 0.336 | 0.187 | 0.038 | 0.003 | 0.0 |
| | ICC | 0.538 | 0.311 | 0.114 | 0.029 | 0.007 | 0.001 |
| | IQ | 0.554 | 0.343 | 0.078 | 0.024 | 0.001 | 0.0 |
| | ITS | 0.469 | 0.306 | 0.137 | 0.045 | 0.032 | 0.011 |
| VARGPT | TCC | 0.707 | 0.241 | 0.044 | 0.006 | 0.002 | 0.0 |
| | ICC | 0.858 | 0.069 | 0.046 | 0.015 | 0.009 | 0.003 |
| | IQ | 0.852 | 0.073 | 0.052 | 0.015 | 0.005 | 0.003 |
| | ITS | 0.832 | 0.054 | 0.057 | 0.036 | 0.018 | 0.003 |
| VILA-U | TCC | 0.823 | 0.151 | 0.017 | 0.005 | 0.003 | 0.001 |
| | ICC | 0.443 | 0.377 | 0.149 | 0.031 | 0.0 | 0.0 |
| | IQ | 0.447 | 0.356 | 0.15 | 0.041 | 0.006 | 0.0 |
| | ITS | 0.678 | 0.165 | 0.104 | 0.034 | 0.013 | 0.006 |

Table 13: Gap proportion between InternVL trained and human scoring.

| Model | Dimension | Score | | | | | |
|---|---|---|---|---|---|---|---|
| | | 0 | 1 | 2 | 3 | 4 | 5 |
| Anole | TCC | 0.871 | 0.106 | 0.023 | 0.0 | 0.0 | 0.0 |
| | ICC | 0.738 | 0.193 | 0.05 | 0.012 | 0.007 | 0.0 |
| | IQ | 0.703 | 0.206 | 0.068 | 0.019 | 0.003 | 0.001 |
| | ITS | 0.773 | 0.161 | 0.052 | 0.013 | 0.001 | 0.0 |
| GPT-4o+DALL-E | TCC | 0.801 | 0.191 | 0.008 | 0.0 | 0.0 | 0.0 |
| | ICC | 0.767 | 0.211 | 0.019 | 0.001 | 0.002 | 0.0 |
| | IQ | 0.777 | 0.205 | 0.014 | 0.001 | 0.002 | 0.001 |
| | ITS | 0.778 | 0.201 | 0.013 | 0.004 | 0.003 | 0.001 |
| DDiT | TCC | 0.967 | 0.013 | 0.012 | 0.007 | 0.001 | 0.0 |
| | ICC | 0.77 | 0.186 | 0.034 | 0.006 | 0.004 | 0.0 |
| | IQ | 0.72 | 0.215 | 0.045 | 0.019 | 0.001 | 0.0 |
| | ITS | 0.962 | 0.015 | 0.014 | 0.007 | 0.001 | 0.001 |
| Emu3 | TCC | 0.775 | 0.2 | 0.021 | 0.004 | 0.0 | 0.0 |
| | ICC | 0.747 | 0.196 | 0.037 | 0.016 | 0.004 | 0.0 |
| | IQ | 0.748 | 0.2 | 0.038 | 0.009 | 0.001 | 0.004 |
| | ITS | 0.741 | 0.185 | 0.039 | 0.031 | 0.004 | 0.0 |
| SEIR | TCC | 0.726 | 0.25 | 0.017 | 0.006 | 0.001 | 0.0 |
| | ICC | 0.816 | 0.159 | 0.022 | 0.002 | 0.0 | 0.001 |
| | IQ | 0.737 | 0.241 | 0.008 | 0.002 | 0.002 | 0.01 |
| | ITS | 0.732 | 0.233 | 0.028 | 0.005 | 0.001 | 0.001 |
| Gemini+Flux | TCC | 0.786 | 0.195 | 0.012 | 0.004 | 0.0 | 0.003 |
| | ICC | 0.728 | 0.243 | 0.02 | 0.007 | 0.002 | 0.0 |
| | IQ | 0.746 | 0.236 | 0.016 | 0.002 | 0.0 | 0.0 |
| | ITS | 0.806 | 0.142 | 0.028 | 0.02 | 0.001 | 0.003 |
| Janus-Pro | TCC | 0.855 | 0.133 | 0.012 | 0.0 | 0.0 | 0.0 |
| | ICC | 0.725 | 0.209 | 0.057 | 0.008 | 0.001 | 0.0 |
| | IQ | 0.699 | 0.226 | 0.052 | 0.019 | 0.004 | 0.0 |
| | ITS | 0.679 | 0.219 | 0.081 | 0.018 | 0.001 | 0.002 |
| Liquid | TCC | 0.846 | 0.138 | 0.012 | 0.004 | 0.0 | 0.0 |
| | ICC | 0.796 | 0.169 | 0.021 | 0.012 | 0.001 | 0.001 |
| | IQ | 0.716 | 0.224 | 0.052 | 0.006 | 0.001 | 0.001 |
| | ITS | 0.71 | 0.205 | 0.066 | 0.017 | 0.002 | 0.0 |
| Show-o | TCC | 0.788 | 0.176 | 0.023 | 0.01 | 0.003 | 0.0 |
| | ICC | 0.743 | 0.205 | 0.028 | 0.019 | 0.004 | 0.001 |
| | IQ | 0.718 | 0.215 | 0.059 | 0.006 | 0.001 | 0.001 |
| | ITS | 0.676 | 0.214 | 0.082 | 0.021 | 0.004 | 0.003 |
| Show-o-Turbo | TCC | 0.718 | 0.213 | 0.054 | 0.013 | 0.002 | 0.0 |
| | ICC | 0.684 | 0.243 | 0.052 | 0.015 | 0.006 | 0.0 |
| | IQ | 0.713 | 0.242 | 0.024 | 0.017 | 0.004 | 0.0 |
| | ITS | 0.715 | 0.214 | 0.06 | 0.01 | 0.001 | 0.0 |
| VARGPT | TCC | 0.797 | 0.15 | 0.047 | 0.006 | 0.0 | 0.0 |
| | ICC | 0.913 | 0.04 | 0.032 | 0.008 | 0.004 | 0.003 |
| | IQ | 0.922 | 0.033 | 0.03 | 0.011 | 0.001 | 0.003 |
| | ITS | 0.917 | 0.027 | 0.038 | 0.016 | 0.001 | 0.001 |
| VILA-U | TCC | 0.906 | 0.078 | 0.012 | 0.002 | 0.002 | 0.0 |
| | ICC | 0.719 | 0.208 | 0.062 | 0.01 | 0.001 | 0.0 |
| | IQ | 0.702 | 0.216 | 0.059 | 0.022 | 0.001 | 0.0 |
| | ITS | 0.827 | 0.122 | 0.044 | 0.005 | 0.001 | 0.001 |

Table 14: Gap proportion between QwenVL_trained and human scoring.

| Model | Dimension | Score | | | | | |
|---|---|---|---|---|---|---|---|
| | | 0 | 1 | 2 | 3 | 4 | 5 |
| Anole | TCC | 0.857 | 0.121 | 0.022 | 0.0 | 0.0 | 0.0 |
| | ICC | 0.713 | 0.225 | 0.052 | 0.006 | 0.004 | 0.0 |
| | IQ | 0.701 | 0.229 | 0.05 | 0.016 | 0.004 | 0.0 |
| | ITS | 0.748 | 0.189 | 0.048 | 0.013 | 0.001 | 0.001 |
| GPT-4o+DALL-E | TCC | 0.783 | 0.207 | 0.01 | 0.0 | 0.0 | 0.0 |
| | ICC | 0.765 | 0.209 | 0.023 | 0.001 | 0.001 | 0.001 |
| | IQ | 0.757 | 0.227 | 0.015 | 0.0 | 0.001 | 0.0 |
| | ITS | 0.773 | 0.187 | 0.029 | 0.009 | 0.002 | 0.0 |
| DDiT | TCC | 0.974 | 0.015 | 0.007 | 0.003 | 0.001 | 0.0 |
| | ICC | 0.732 | 0.223 | 0.039 | 0.005 | 0.001 | 0.0 |
| | IQ | 0.733 | 0.227 | 0.033 | 0.004 | 0.003 | 0.0 |
| | ITS | 0.969 | 0.013 | 0.013 | 0.004 | 0.001 | 0.0 |
| Emu3 | TCC | 0.757 | 0.22 | 0.021 | 0.002 | 0.0 | 0.0 |
| | ICC | 0.729 | 0.233 | 0.019 | 0.019 | 0.0 | 0.0 |
| | IQ | 0.749 | 0.224 | 0.021 | 0.005 | 0.001 | 0.0 |
| | ITS | 0.781 | 0.17 | 0.04 | 0.007 | 0.001 | 0.001 |
| SEIR | TCC | 0.699 | 0.279 | 0.018 | 0.003 | 0.001 | 0.0 |
| | ICC | 0.659 | 0.309 | 0.022 | 0.01 | 0.0 | 0.0 |
| | IQ | 0.704 | 0.281 | 0.014 | 0.001 | 0.0 | 0.0 |
| | ITS | 0.707 | 0.268 | 0.02 | 0.004 | 0.001 | 0.0 |
| Gemini+Flux | TCC | 0.735 | 0.236 | 0.022 | 0.004 | 0.003 | 0.0 |
| | ICC | 0.708 | 0.262 | 0.023 | 0.004 | 0.003 | 0.0 |
| | IQ | 0.712 | 0.263 | 0.024 | 0.001 | 0.0 | 0.0 |
| | ITS | 0.703 | 0.271 | 0.015 | 0.01 | 0.001 | 0.0 |
| Janus-Pro | TCC | 0.841 | 0.147 | 0.011 | 0.001 | 0.0 | 0.0 |
| | ICC | 0.672 | 0.262 | 0.054 | 0.012 | 0.0 | 0.0 |
| | IQ | 0.667 | 0.252 | 0.072 | 0.007 | 0.001 | 0.001 |
| | ITS | 0.7 | 0.224 | 0.064 | 0.01 | 0.001 | 0.001 |
| Liquid | TCC | 0.833 | 0.152 | 0.011 | 0.004 | 0.0 | 0.0 |
| | ICC | 0.704 | 0.245 | 0.03 | 0.018 | 0.002 | 0.001 |
| | IQ | 0.699 | 0.244 | 0.044 | 0.01 | 0.003 | 0.0 |
| | ITS | 0.661 | 0.269 | 0.06 | 0.009 | 0.001 | 0.0 |
| Show-o | TCC | 0.774 | 0.195 | 0.021 | 0.01 | 0.0 | 0.0 |
| | ICC | 0.688 | 0.247 | 0.043 | 0.022 | 0.0 | 0.0 |
| | IQ | 0.722 | 0.248 | 0.023 | 0.004 | 0.002 | 0.001 |
| | ITS | 0.693 | 0.244 | 0.051 | 0.01 | 0.001 | 0.001 |
| Show-o-Turbo | TCC | 0.69 | 0.246 | 0.047 | 0.016 | 0.001 | 0.0 |
| | ICC | 0.672 | 0.265 | 0.039 | 0.023 | 0.001 | 0.0 |
| | IQ | 0.624 | 0.299 | 0.068 | 0.008 | 0.001 | 0.0 |
| | ITS | 0.747 | 0.197 | 0.045 | 0.009 | 0.001 | 0.001 |
| VARGPT | TCC | 0.82 | 0.159 | 0.017 | 0.004 | 0.0 | 0.0 |
| | ICC | 0.913 | 0.047 | 0.031 | 0.009 | 0.0 | 0.0 |
| | IQ | 0.922 | 0.05 | 0.027 | 0.001 | 0.0 | 0.0 |
| | ITS | 0.925 | 0.038 | 0.019 | 0.016 | 0.001 | 0.001 |
| VILA-U | TCC | 0.89 | 0.097 | 0.012 | 0.001 | 0.0 | 0.0 |
| | ICC | 0.639 | 0.277 | 0.058 | 0.025 | 0.001 | 0.0 |
| | IQ | 0.646 | 0.272 | 0.072 | 0.01 | 0.0 | 0.0 |
| | ITS | 0.8 | 0.127 | 0.06 | 0.011 | 0.002 | 0.0 |

Table 15: Evaluation accuracy A@1 comparison across judges

| Model | Dim. | GPT-4o | InternVL | InternVL_trained | QwenVL | QwenVL_trained |
|---|---|---|---|---|---|---|
| Anole | TCC | 0.805 | 0.816 | 0.977 | 0.972 | 0.977 |
| | ICC | 0.848 | 0.859 | 0.930 | 0.816 | 0.937 |
| | IQ | 0.828 | 0.776 | 0.908 | 0.870 | 0.929 |
| | ITS | 0.823 | 0.851 | 0.933 | 0.779 | 0.936 |
| GPT-4o+DALL-E | TCC | 0.971 | 0.983 | 0.991 | 0.978 | 0.990 |
| | ICC | 0.961 | 0.956 | 0.977 | 0.949 | 0.974 |
| | IQ | 0.931 | 0.916 | 0.981 | 0.903 | 0.983 |
| | ITS | 0.938 | 0.933 | 0.978 | 0.866 | 0.959 |
| DDiT | TCC | 0.960 | 0.963 | 0.979 | 0.967 | 0.989 |
| | ICC | 0.846 | 0.901 | 0.955 | 0.850 | 0.954 |
| | IQ | 0.837 | 0.831 | 0.934 | 0.801 | 0.958 |
| | ITS | 0.885 | 0.951 | 0.976 | 0.958 | 0.982 |
| Emu3 | TCC | 0.896 | 0.915 | 0.974 | 0.944 | 0.975 |
| | ICC | 0.867 | 0.899 | 0.942 | 0.871 | 0.961 |
| | IQ | 0.912 | 0.858 | 0.946 | 0.924 | 0.972 |
| | ITS | 0.811 | 0.846 | 0.925 | 0.819 | 0.950 |
| SEIR | TCC | 0.917 | 0.940 | 0.975 | 0.912 | 0.978 |
| | ICC | 0.936 | 0.946 | 0.973 | 0.925 | 0.967 |
| | IQ | 0.945 | 0.937 | 0.976 | 0.930 | 0.983 |
| | ITS | 0.876 | 0.881 | 0.964 | 0.848 | 0.974 |
| Gemini+Flux | TCC | 0.946 | 0.960 | 0.980 | 0.961 | 0.970 |
| | ICC | 0.941 | 0.945 | 0.970 | 0.905 | 0.969 |
| | IQ | 0.950 | 0.941 | 0.981 | 0.930 | 0.975 |
| | ITS | 0.924 | 0.911 | 0.946 | 0.936 | 0.973 |
| Janus-Pro | TCC | 0.727 | 0.716 | 0.987 | 0.775 | 0.987 |
| | ICC | 0.806 | 0.846 | 0.932 | 0.743 | 0.933 |
| | IQ | 0.796 | 0.687 | 0.924 | 0.778 | 0.918 |
| | ITS | 0.775 | 0.783 | 0.897 | 0.777 | 0.924 |
| Liquid | TCC | 0.853 | 0.827 | 0.983 | 0.918 | 0.984 |
| | ICC | 0.880 | 0.890 | 0.964 | 0.881 | 0.948 |
| | IQ | 0.877 | 0.820 | 0.939 | 0.939 | 0.942 |
| | ITS | 0.770 | 0.775 | 0.914 | 0.716 | 0.929 |
| Show-o | TCC | 0.867 | 0.905 | 0.963 | 0.891 | 0.969 |
| | ICC | 0.842 | 0.874 | 0.947 | 0.852 | 0.935 |
| | IQ | 0.827 | 0.781 | 0.932 | 0.820 | 0.969 |
| | ITS | 0.801 | 0.808 | 0.889 | 0.829 | 0.936 |
| Show-o-Turbo | TCC | 0.771 | 0.861 | 0.929 | 0.772 | 0.935 |
| | ICC | 0.861 | 0.895 | 0.926 | 0.848 | 0.936 |
| | IQ | 0.815 | 0.770 | 0.954 | 0.897 | 0.922 |
| | ITS | 0.816 | 0.838 | 0.928 | 0.775 | 0.943 |
| VARGPT | TCC | 0.706 | 0.638 | 0.946 | 0.948 | 0.978 |
| | ICC | 0.924 | 0.924 | 0.953 | 0.926 | 0.959 |
| | IQ | 0.956 | 0.939 | 0.953 | 0.924 | 0.971 |
| | ITS | 0.934 | 0.922 | 0.944 | 0.884 | 0.962 |
| VILA-U | TCC | 0.865 | 0.912 | 0.982 | 0.973 | 0.986 |
| | ICC | 0.832 | 0.868 | 0.926 | 0.818 | 0.915 |
| | IQ | 0.772 | 0.711 | 0.916 | 0.802 | 0.917 |
| | ITS | 0.815 | 0.879 | 0.948 | 0.842 | 0.926 |
| Average | A@1 | 0.865 | 0.866 | 0.945 | 0.875 | 0.954 |

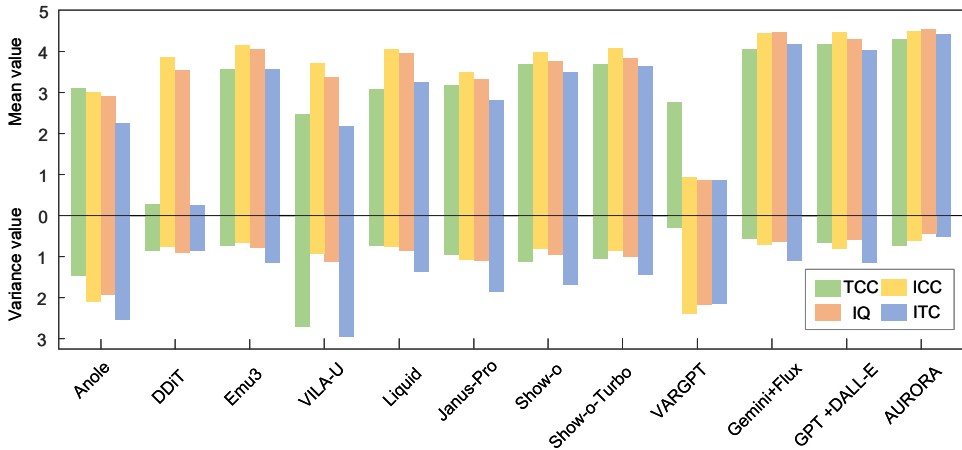

Figure 7: A visualization of mean and variance of different generators

## D.8 Detailed Analysis of the Capabilities of Different Generators

**Mean and Variance Analysis.** Figure 7 presents the mean and variance of evaluation scores across TCC, ICC, IQ, and ITS for different generators. Several important observations emerge from the results. First, DDiT exhibits the lowest mean scores in both TCC (0.38) and ITS (0.37), indicating poor content coverage and weak image-text synergy. Second, VARGPT shows mean scores below 1 across ICC, IQ, and ITS, suggesting significant deficiencies in visual generation capabilities and multimodal alignment. Third, VILA-U demonstrates the highest variance in TCC and ITS among all models, implying that its performance is highly unstable across different questions. In contrast, Gemini+Flux and GPT-4o+DALL-E achieve mean scores above 4.0 across all evaluation dimensions, reflecting generally strong performance. However, their relatively high variance in ITS reveals that they still struggle with maintaining image-text consistency and complementarity across samples. Most notably, SEIR method consistently outperforms all other generators across all four dimensions, achieving the highest mean scores while maintaining the lowest variance. This indicates not only superior quality but also high stability and robustness in both textual and visual generation. These findings collectively highlight the importance of both quality and consistency for robust multimodal generation, and demonstrate the synergy of SEIR in constructing high quality, stable datasets.

## E Question Templates and Topic Hierarchy

In our framework, **question templates** are designed to model the style of human queries rather than to encode domain knowledge. Specifically, these templates capture the recurrent **syntactic and pragmatic structures** through which users naturally formulate requests. For example, variations such as *"Please recommend some equipment needed for hiking,"*, *"Tell me some equipment needed for hiking,"* or *"Could you tell me what equipment is needed for hiking?"* all convey the same underlying intent but differ in their linguistic form. By systematically enumerating such structures, we obtain broad coverage of syntactic patterns for conversational queries (e.g., "can you . . . ," "please . . . "), while leaving the semantic content to be drawn from a large topic hierarchy. This separation ensures that the **linguistic diversity** of user queries can be effectively modeled: the question template specifies *how* a request is asked, whereas the the 3500-topic hierarchy defines *what* the request is about. The combination of these two components enables our dataset to achieve high diversity while faithfully reflecting real-world user interaction styles.

### E.1 Question Templates

```
Do you know ***? Can you draw a picture of it for me?
Do you know what *** looks like? Please draw an image of it for me.
```

```
I'm very interested in ***. Please help me describe it and draw a
portrait of it.
Can you imagine what *** looks like? Please draw a portrait of it for me.
Have you paid attention to ***? Can you tell me something about it?
Besides, can you depict it in a painting?
What do you think *** looks like? Can you draw what this place might look
 like?
What does *** look like? Can you draw a sketch of it for me?
Do you know about ***? Please draw an image of them for me.
I heard that *** is very attractive. Can you introduce it to me? Then
draw an image of it for me.
I need a painting of *** now. Please help me describe it and draw it.
I'm paying a lot of attention to *** now. Do you know it? By the way,
help me draw a picture to introduce it.
Can you draw a picture of ***? Besides, can you give me some science
knowledge about ***?
Have you ever seen the scene of ***? What does *** usually look like?
Please draw a picture for me.
Will there be ** in the ** of **? Can you draw a picture of this scene
for me?
What is the most wonderful *** you have ever seen? Please draw a scene
picture of *** for me.
What kind of wonders can be seen in ***? I'm really curious about the
scene. Can you show it to me?
Can you imagine ***, with ** and **, creating a moment of ** and **? Help
 me draw them.
Can you describe *** for me? It would be even better if there is a
painting.
Please introduce *** and draw about it.
Hey, can you tell me *** and draw a picture?
...
```

### E.2 TOPIC HIERARCHY

To provide a clearer picture of the dataset's composition, the table 16 below shows the distribution across our 8 primary domains, including the number of categories and fine-grained topics within each. As you can see, the data is well-distributed across diverse topics like Natural Scenery (19.88%), Cultural Scenery (15.87%), and Animals (14.77%), ensuring broad and deep coverage.

Table 16: Data distribution of InterSyn across 8 primary domains. The table details the percentage domain distribution (Domain Dist.), distribution per Category (Dist. per Cat.), number of categories (# Categories), and number of topics for each domain (# Topics), demonstrating broad and deep coverage.

| Domain | Domain Dist. (%) | Dist. per Cat. (%) | # Categories | # Topics |
|---|---|---|---|---|
| Animals | 14.77 | 1.64 | 9 | 517 |
| Plants | 10.13 | 1.27 | 8 | 355 |
| Natural Scenery | 19.88 | 2.48 | 8 | 695 |
| Cultural Scenery | 15.87 | 1.98 | 8 | 555 |
| Objects | 10.65 | 1.33 | 8 | 373 |
| Activities | 11.20 | 1.40 | 8 | 392 |
| Food | 6.71 | 0.84 | 8 | 235 |
| Culture | 10.80 | 1.35 | 8 | 378 |

```
{
"Animals": {
"Terrestrial Animals": ["Giant panda", "Snow leopard", "Black bear", "Red
 panda", "Tibetan antelope", "Argali"...],
"Marine Animals": ["Blue whale", "Killer whale", "Great white shark", "
Humpback whale", "Dolphin", "Octopus"...],
```

```
"Extinct Animals": ["Dinosaur", "Dodo", "Woolly mammoth", "Saber-toothed
tiger", "Pterosaur", "Stegosaurus"...],
"Domesticated Animals": ["Pet cat", "Pet dog", "Pet bird", "Mouse", "
Ornamental fish", "Cow", "Sheep", "Pig"...]
},
"Plants": {
"Edible Plants": ["Rice", "Wheat", "Corn", "Sorghum", "Oat", "Buckwheat",
 "Quinoa", "Millet", "Barley"...],
"Medicinal Plants": ["Ginseng", "Wolfberry", "Coptis chinensis", "
Notoginseng", "Astragalus membranaceus", "Angelica sinensis"...],
"Ornamental Plants": ["Rose", "Tulip", "Chrysanthemum", "Peony", "Chinese
 peony", "Lily", "Narcissus", "Hyacinth", "Iris"...]
},
"Natural Scenery": {
"Mountains & Forests": ["The flag cloud of Mount Everest in the Himalayas
", "Alpine meadows and wildflowers in the Alps"...],
"Water & Weather": ["Hawaiian volcanic lava flowing into the sea", "The
blue-domed church in Santorini, Greece", "A dugout canoe in the lagoon of
 Tahiti"...],
"Deserts & Volcanoes": ["The sharp ridges on the backlit side of sand
dunes", "The winding silhouette of a camel caravan"...],
"Seasons & Landforms": ["Red-crowned cranes dancing in the winter snow in
 Hokkaido, Japan", "The tulip maze in Keukenhof Gardens in the
Netherlands in spring"...]
},
"Cultural Scenery": {
"Cities & Villages": ["The mirror-like water surface of the terraced
fields in Yuanyang, Yunnan", "The lavender fields in Provence, France
"...],
"Religion & Religious Sites": ["Devout believers praying in front of the
Western Wall in Jerusalem", "The play of light and shadow under the dome
of St..."],
"Heritages & Wonders": ["The giant paintings of the Nazca Lines in Peru
seen from above", "The Treasury at the end of the Siq in Petra, Jordan",
"The Pyramids of the Sun and Moon in Teotihuacan,..."]
},
"Objects": {
"Household & Daily Items": ["Bench", "Chair", "Sofa", "Coffee table", "
Bookshelf", "Wardrobe", "Desk", "Dressing table", "Bed", "Dining table",
"Dining chair"...],
"Military & Security": ["Pistol", "Rifle", "Submachine gun", "Machine gun
", "Artillery", "Missile", "Tank", "Armored vehicle", "Fighter jet"...],
"Tools & Equipment": ["Fire hydrant", "Wrench", "Screw", "Hammer", "
Shovel", "Screwdriver", "Tape measure", "Electric drill", "Pliers", "Saw
", "File", "Soldering iron"...],
"Energy & Industry": ["Generator", "Solar panel", "Wind turbine", "
Hydraulic generator", "Battery", "Inverter", "Transformer", "Charging
pile", "Oil drum", "Gas cylinder"...],
"Transportation & Communication": ["Bicycle", "Car", "Motorcycle", "
Airplane", "Bus", "Train", "Truck", "Ship", "Traffic light", "Tricycle",
"Electric scooter"...]
},
"Activities": {
"Daily Life & Occupations": ["Doctor", "Firefighter", "Farmer", "Teacher
", "Lawyer", "Craftsman", "Researcher", "Photographer", "Singer", "Dancer
", "Painter", "Journalist"....],
"Emotional &Social Interactions": ["Hug", "Kiss", "Meet", "Talk", "
Lecture", "Study", "Shake hands", "Comfort", "Celebrate", "Take a group
photo", "Quarrel", "Share", "Wave"...],
"Sports & Labor": ["Run", "Play basketball", "Play football", "Play
volleyball", "Play badminton", "Play tennis", "Play table tennis", "Jump
", "Ride a bike", "Box", "Wrestle"...]
},
"Food": {
```

```
"Regional Cuisines": ["Mapo Tofu", "Ramen", "Braised Pork Belly in Soy
Sauce", "Scrambled Eggs with Tomatoes", "Shredded Pork with Green Peppers
", "Braised Beef with Potatoes"...],
"Baked Goods & Desserts": ["Caramel Pudding", "Macaron", "Donut", "Cake",
 "Yogurt", "French Croissant", "Italian Tiramisu", "German Black Forest
Cake", "Japanese Wagashi"...],
"Processed Foods": ["Snacks", "Canned Food", "Frozen Food", "Biscuits", "
Chocolate Biscuits", "Ice Cream", "Popcorn", "Potato Chips", "Canned Fish
", "Frozen Dumplings"...],
"Beverages": ["Red Wine", "Chinese Baijiu", "Beer", "Coke", "Juice", "Tea
", "Milk", "Soda Water", "French Champagne", "Italian Espresso", "
Japanese Sake", "Korean Makgeolli"...],
"Pet Food": ["Dog Food", "Cat Food", "Chew Sticks", "Bones", "Pet Canned
Food", "Freeze-Dried Chicken Pieces", "Salmon-Flavored Cat Treats"...]
},
"Culture": {
"Material Culture": ["Hanfu (Han Chinese Clothing)", "Qipao (Cheongsam)",
 "Kimono", "Indian Sari", "Western Suit", "Wedding Dress", "Tangzhuang (
Tang-style Costume)", "Mongolian Robe"...],
"Spiritual Culture": ["The Dragon Totem in Ancient China", "The Phoenix
Totem", "The Eagle Totem of the Native Americans", "The Wolf Totem", "The
 Rainbow Serpent Totem of the Australian Aborigines"...],
"Behavioral Culture": ["Traditional Chinese Wedding", "Western Church
Wedding", "Coming-of-Age Ceremony", "Crowning Ceremony", "Sacrificial
Ceremony", "Japanese Tea Ceremony Etiquette", "The Hongi (Nose Rubbing)
of the Maori People", "The Namaste of India", "The Apprenticeship
Ceremony in Thailand", "The Torch Festival Ceremony of the Yi Ethnic
Group"...]
}
}
```

# F    EVALUATION DIMENSIONS FOR DATASET QUALITY

## F.1    QUESTION EVALUATION DIMENSIONS

**Reasonableness of Expression**: The question statement is smooth, without any grammatical errors, and the words are used accurately and appropriately. For example, "Please introduce the Great Wall to me and also give me a picture of the Great Wall" is a reasonable expression; while "Tell me about the Great Wall, and give me a picture" has a problem of confused expression. Such questions will affect the model's understanding of the intention and make it difficult to give an accurate answer.

**Clarity of Requirements**: Clearly indicate that the model is required to provide both text and image responses simultaneously. For instance, "Introduce the appearance characteristics of Notre-Dame de Paris and provide a high-definition frontal picture", which clearly puts forward the dual requirements of text description and image acquisition; if the question is just "What does Notre-Dame de Paris look like", without clearly stating the image requirement, it does not meet the requirements and cannot effectively guide the model to give a comprehensive response.

**Focus of the Theme**: The question revolves around a single and clear theme and will not jump between multiple unrelated themes. For example, "Introduce the geographical features of Mount Fuji and attach a distant view of Mount Fuji", with the theme focused on Mount Fuji; while "Tell me about Mount Fuji and then talk about the Eiffel Tower, and give two corresponding pictures", which involves two different themes, may lead to unclear logical answers from the model and is not conducive to the standardized construction of the dataset.

**Feasibility and Clarity**: Based on common sense judgment, the content involved in the question is something that the model has the ability to answer through language and images, and there is no way of multiple interpretations, and the model can accurately grasp the questioner's intention. For example, "Describe the living habits of giant pandas and give a picture of a panda eating bamboo", the model can answer based on its existing knowledge reserve and image generation ability, and the intention is clear; however, "Tell me what it's like for a person to take a bath in volcanic magma and give a picture", such questions seriously deviate from reality and lack scientific basis. The model can

neither answer based on existing knowledge, nor is there a real-world reference for image generation, which will lead to absurd and meaningless generation results and greatly reduce the reliability and practicality of the dataset.

**Appropriateness of Length**: The length of the question is moderate, which not only contains enough key information to guide the model to generate high quality answers but also is not too long and complicated for the model to grasp the key points. Generally speaking, short and concise questions are helpful for the model to quickly understand the intention, such as "Introduce the Forbidden City and give a panoramic picture"; but being too short may lack sufficient information, such as "Forbidden City, picture"; and overly long and cumbersome questions, like "Please introduce in detail the process of the changes of the Forbidden City since its construction in the Ming Dynasty through various dynasties, including the evolution of architectural styles, the transformation of functional uses and other aspects, and provide a high-definition panoramic picture that can comprehensively display the current overall layout of the Forbidden City. At the same time, ensure that the picture contains the main palaces, courtyards, city walls and other iconic elements of the Forbidden City", may cause confusion when the model processes it. The ideal length can be determined according to practical experience and testing. Usually, about 15 - 50 words is more appropriate, which can convey the requirements completely and also facilitate the model to process efficiently.

## F.2 INTERLEAVED IMAGE-TEXT ANSWER EVALUATION DIMENSIONS

**Text Content Completeness (0-5 points)**: This dimension only focuses on the correspondence between the text response and the question, whether the content precisely matches the user's needs, and whether the information is complete and error-free. It does not consider the output of any other dimensions and evaluation criteria.
0 points: No text appears; 1 point: The text answer has nothing to do with the question; it is completely wrong, completely divorced from the question, and there is no positive response to the text requirement; there is less content but there are truncations and random spitting characters. 2 points: The text answer can only cover a small part of the elements required in the question, and there is a large amount of unreasonable content; there is a very obvious phenomenon of text truncation that seriously affects the original information; the content is very long or very short, which seriously affects the reading. 3 points: The answer can correspond to key elements, there is a small amount of unreasonable content, and there may be omissions of key information; the content is too long or too short, but the information basically corresponds. 4 points: The required elements of the question are basically all corresponding, there is no unreasonable content, there is a omission of key information, or the answer is awkward; the content is slightly longer or shorter, but the answer is very correct. 5 points: The content of the answer exactly corresponds to the question, there is no unreasonable content, and the answer is smooth and fluent, with full content.

**Image Content Completeness (0-5 points)**: This dimension only focuses on the correspondence between the image content and the question (considering the content of the picture, the degree to which the image content answers the question). Whether the key parts are retained, and whether there is an obvious lack of objects.
0 points: No image appears; 1 point: The content of the image is completely wrong, and no key elements are depicted at all; the image has no connection to the problem, even if the image itself is of good quality. 2 points: About half of the key elements required for the problem are missing, and there are a large number of unreasonable elements; the elements in the figure may have some connection to the problem, but it is almost impossible to identify what they are. 3 points: Only a small number of key elements required for the problem are missing in the figure, most of the elements can be fully identified, and there are only a few unreasonable content. 4 points: Basically lack the elements required for the problem, and there may be minor flaws in some details. 5 points: All the elements required for the question are completely corresponding, the main body is intact, and the picture content answers the question very well.

**Image Quality (0-5 points)**: This dimension only focuses on the performance of the basic generation technology of the image (do not consider the content of the picture). Whether it is clear, whether there are blurred, noisy or out-of-focus areas, truncations or damages (that is, the judgment of image aesthetics and subjective quality).
0 points: No picture; 1 point: The image is very ugly, and it is almost impossible to identify the image content. 2 points: The image looks ugly, the overall image is blurred but can be barely recog-

nized; 3 points: The image is medium in appearance, and the main elements can be distinguished, but other elements are blurred. 4 points: The image looks good, the picture is relatively clear, and there is no visible blurring phenomenon; 5 points: The image looks good, the details are sharp without blur, and the image quality is very high.

**Image-Text Synergy (0–5 points)**: This dimension evaluates the degree of alignment and complementarity between the textual and visual components of a response. It focuses not only on how well the entities or scenes described in the text are accurately and completely depicted in the image, but also on whether the text and image together form a coherent and mutually supportive answer to the question.
0 points: The image and text are completely unrelated. Additionally, if either the image or the text is missing (i.e., "null"), the response is assigned 0 points. 1 point: The image and text are minimally related, with only a few elements weakly corresponding. The response lacks coherence and fails to effectively address the question. 2 points: Around half of the key elements described in the text are reflected in the image, but significant mismatches remain. The overall synergy is poor. 3 points: Most elements between the text and image are consistent, but a few important mismatches or omissions in key entities or scenes reduce the completeness of the response. 4 points: Nearly all elements between the text and image are consistent, with only minor mismatches in non-critical details. The response answers the question well, but there may be redundancy between the two modalities, limiting their complementarity. 5 points: The text and image are perfectly aligned, with all described elements accurately and fully presented. The two modalities work together in a complementary way to form a complete and informative response without unnecessary duplication.

# G  ALL PROMPTS USED IN THIS WORK

## G.1  PROMPTS USED IN SEIR METHOD

Only a simple example of a single-round dialogue generation prompt is provided here. The most detailed prompts are given in detail in the open-source code. Detailed prompts can be found in our code.

Here is the prompt for the question generation:

```
I am building a question-answer dataset.
The topic of this dataset item is ({topic}). Your task is to generate a
question based on this topic.
The length of the question should not exceed 50 words. Here is the
question template: \n{ques_temp}.\n
The new question you generate can refer to the sentence pattern of the
question template.
The question must meet the following detailed requirements:
1. **Incorporate Image Request Naturally**: Clearly express the need to
generate a picture, but use varied and creative expressions to make the
request feel natural and human-like. Avoid repetitive phrases like 'maybe
 generate a picture.' Instead, use diverse sentence structures to request
 the image.
2. **Varied Sentence Structures**: Diversify how questions are phrased.
Use different ways of asking, such as open-ended questions, hypothetical
scenarios, or requests for examples.
3. **Conciseness and Clarity**: Ensure the question is still concise and
immediately understandable but without sounding repetitive or formulaic.
Avoid redundant language.
4. **Topic Relevance**: Keep the question focused on the given topic ({
topic}), ensuring it remains engaging and meaningful. Avoid weak
connections to the topic.
5. **Approachable Tone**: Use a conversational, approachable tone that
mimics real human interactions. Keep it friendly and engaging, avoiding
overly formal or robotic expressions.
6. **Lexical Simplicity with Creativity**: Use everyday vocabulary with
occasional creative language that fits the topic. Ensure accessibility
for a broad audience while maintaining interest.
```

```
7. **Question Value and Inspiration**: Make the question thought-
provoking or creative, capable of inspiring meaningful answers. Avoid
overly simple or overly complex questions.
8. **Image Context**: Clearly specify what kind of picture is expected,
but do so creatively.
Output only the generated question directly. Do not include explanations,
 instructions, or any extra text.
```

Here is the prompt to get the the suggestions for the question:

```
I am currently constructing a question-answer dataset. The first step is
to imitate human needs and tone based on a certain topic and ask a
question.
This question needs to include the requirement for generating textual
content and a picture.
The topic is: ({topic}).\n
The following is a question generated based on this topic:\n{old_q}\n
You need to analyze the quality of this question from a human perspective
, such as whether the question is too wordy?
Is the question sentence pattern not commonly used in human daily
communication? How well does the question fit the topic?
Does the tone of the question sound human? Are there any uncommon
expressions in the sentence?
Is it a meaningless question? Does the question contain a request for
generating an image? Is the generated question easy to answer? And so on.
You need to help me provide revision suggestions. It would be best if the
 suggestions are concise and brief, and not too long.
If you think the original question is not good in other aspects, you need
 to help me give modification suggestions.
Only output the modification suggestions in the end, and there is no need
 to output the modified results.
Your output should conform to this format {json_format}
If you think the original question is good enough, you don't need to give
 improvement suggestions. You only need to output None.
Therefore, your final output is either None or the modification
suggestions.
```

Here is the prompt for the question modification:

```
I am currently constructing a question-answer dataset.\n
The following is the original question generated by an LLM:\n{old_q}\n\n
However, I believe the quality of this question can be improved, as it
doesn't sound like something people would naturally ask in daily
communication.\n\n
I have provided some modification suggestions: {mod_q_suggestion}.\n
Please revise the question based on these suggestions and the given topic
, making it sound more natural and human-like.\n
Finally, output only the modified question without any additional text.
```

Here is the prompt to get the answer of the question:

```
Currently, I'm constructing a question-answer dataset. This is the
current question: \n{final_q}\n
Since this question usually contains a requirement for textual answer and
 image generation., but you don't need to generate the actual image.
Instead, you should generate an answer and a description of the image
according to the question.
To ensure high Image-Text Synergy (ITS), write the answer line so it
gives the core explanation while referencing key visual elements, and
write the caption line so it adds complementary details that the text
omits; the two lines must stay tightly aligned, avoid duplication, and
together convey more than either could alone.
Therefore, your response should include an answer to the question: answer
; and a description of the image: caption. And you are not allowed to
```

```
output responses like 'I can't generate images.' You need to pretend that
 you can.
The image description must not exceed 65 words. This last point is very
important! You just need to output in two lines and there should be no
other content. The output content: start the first line with 'answer:',
representing the answer; start the second line with 'caption:',
representing the caption.
Your answer should be related to the previous content and must not be
repetitive.
```

Here is the prompt for the suggestions for answer modification:

```
Currently, I'm constructing a question-answer dataset.
Here is the question: \n{final_q}\n.
The question usually includes the requirement for textual answer and
image generation.
Then, here is the answer to this question:\n{old_ac}\n
The answer is divided into two parts, including the textual answer to the
 question and an image description.
Do you think the combination of this answer and image description can
fully meet the requirements of the question? Are the image description
and the answer content consistent and not redundant?
How is the correlation among the question, the answer and the image
description? Does it conform to the habits of human answering questions?
If you were a knowledgeable human expert, how do you think you would
answer this question? Would the answer seem too wordy?
Would the overlap between the answer and the image description be too
high? Can the image description well summarize a picture?
If you were a nitpicking critic, do you think there are areas for
improvement in this question, the answer and the image description?
Would the image description be too short and not rich enough in content?
Are there any discriminatory elements in the answer and the image
description? And so on.
You can give modification suggestions based on the above aspects. Or if
you think the answer is unreasonable in other aspects, you also need to
give your modification suggestions.
In addition, the modification suggestions need to be divided into two
parts: the answer and the image description. And the content needs to be
concise and condensed, not overly long.
Or if you think the answer and the image description are already perfect,
 you don't need to put forward improvement suggestions, and just output
None.
Therefore, your final output is either None or the modification
suggestions.
Only output the modification suggestions in the end, and there is no need
 to output the modified results.Your output should conform to this format
 {json_format} or None.
```

Here is the prompt for the answer modification:

```
You are tasked with improving the output of a model output based on the
suggestion feedback.
Here is the context and what you need to do step by step:\n\n
Model Output to Modify (old_ac): \n{old_ac}\n
This is the current answer generated by the model. The answer is divided
into two parts:\n
- 'answer': This is the text answer to the question.\n
- 'caption': This is the image description associated with the answer.\n\
n
Modification Suggestion (mod_ac_suggestion): \n{mod_ac_suggestion}\n
This is the suggestion for improving the model's output, including
corrections or enhancements to both the 'answer' and 'caption' parts.\n\n
Your task is to:\n
- According to the provided mod_ac_suggestion, update the 'answer' and '
caption' sections in old_ac.\n
```

```
- Ensure that the updated 'caption' does not exceed 65 words.\n
- Follow the specified format strictly.\n\n
Important: You just need to output in two lines and there should be no
other content.
The output content: start the first line with 'answer:', representing the
 answer; start the second line with 'caption:', representing the caption.
```

Here is the prompt for the suggestions for caption modification:

```
Currently, I'm constructing a question-answer dataset.
Here is the question: \n{final_q}\n.
This question usually contains a request for generating textual content
and a picture.
Then, this is the original answer final_a: {final_a} and the image
description old_c: {old_c} generated according to this question,
You now need to evaluate the quality of the image description and the
image based on the question and the answer. Does the image match the
image description?
When proposing revisions, follow the Image-Text Synergy (ITS) principle:
suggest changes that make the picture (and its caption) complement rather
 than repeat the fixed textual answer, depict the visual elements the
answer references, reduce redundancy or irrelevant details, and keep full
 factual consistency so that image+text together convey more than either
could alone.
How is the degree of correlation between the image description and the
content of the answer to the question? Can the image description well
summarize the content of the picture?
Are there any unreasonable objects or behaviors in the image? Is the
image description clear and not wordy? And so on.
You can give modification suggestions regarding the image description
based on the above aspects. Suggestions in other aspects not mentioned
above are also highly encouraged to be put forward.
The revision suggestions you provide need to be concise and condensed,
and shouldn't be too long.
If you think the image description and the image for this question and
answer are already perfect, then you don't need to put forward any
suggestions and just output None.
Therefore, your final output is only None or the modification suggestions
.
Only output the modification suggestions in the end, and there is no need
 to output the modified results.Your output should conform to this format
 {json_format} or None.
```

Here is the prompt for the caption modification:

```
Currently, I'm constructing a question-answer dataset. The question
usually includes the requirement for textual answer and image generation.
Then, this is the image description of the answer: \n{old_c}\n
Then I think the quality of the image description is not very high.
I have provided some modification suggestions here: \n{mod_c_suggestion}\
n
Please regenerate the image description according to these suggestions.
The length of the picture description should not exceed 65 words. In the
end, you only need to output the modified image description.
```

## G.2 INTERLEAVED IMAGE-TEXT ANSWER EVALUATION PROMPT USED BY MLLM

Here is the prompt for evaluating the interleaved image-text answer:

```
You are an experienced, fair and impartial judge. Next, I will provide
you with a conversation where a human interacts with different GPTs on
daily topics. In this scenario, the human will pose a text question, and
the GPT's response is based on this question. This response usually
includes a piece of text and image information, but there may be
```

```
exceptions where there is only text or only image information. Now you
need to reasonably rate the response given by the GPT. <chatbegin>
represents the start of the Q&A data, and <chatend> represents the end of
 the Q&A data. The rating of the response is divided into the following
four dimensions, and you should rate the response fairly and impartially
according to the criteria of each dimension.

Here are the four dimensions for evaluating the response:
"""
<Interleaved Image-Text Answer Evaluation Dimensions>
"""
The content of your output rating must strictly conform to the following
format:
[Text Content Completeness: *; Image Content Completeness: *; Image
Quality: *; Image-Text Synergy: *]
your score * for different dimensions, only as a score in (0, 1, 2, 3, 4,
 5). You need to strictly conduct the grading.
Here is the data you need to evaluate, and you need to evaluate the
quality of the Answer from the above four dimensions (both text and image
 may be "null", and the fact that one of them is "null" will not affect
the rating of other dimensions.):
```

## H  BENCHMARK SAMPLES

The partly benchmark examples obtained after modification based on the questions raised by the participants are as follows:

```
Are there fireflies in the forest on a summer night? Can you draw a
picture of this scene for me?
What could a wonderful concert scene be like? Please draw a scene of a
concert for me.
What kind of wonders can be seen in the forest on a cold winter night? I'
m very curious about what that scene would be like. Can you show it to me
?
Who is the king of the African savanna? Can you draw a picture to depict
it?
Can you imagine a serene ocean scene with a setting sun and some seagulls
, creating a calm and relaxing moment? Please draw it for me.
Can you describe a forest for me? It would be even better if there is a
painting.
Hey, can you tell me a really terrifying legend and draw a vivid picture
of it?
Can you describe the scenes in a futuristic music video? If possible, can
 you quickly draw a sketch? I'm really eager to see your ideas!
Hey! What is the daily life of people in the military usually like? Also,
 can you show me what a soldier in military uniform looks like?
Describe the scene of a huge lightning bolt during a storm. Draw a
picture of this scene.
Can you imagine how the concept of the Tree of Life is presented in
different religions? Perhaps a painting showing its symbolism would be
helpful.
Can you describe what the snowy scene in a blizzard is like? I want to
see such a landscape.
Introduce a delicious snack. Describe its appearance, ingredients, and
what makes it so appealing. Also, draw what it looks like.
Describe the traditional decorations of the Lantern Festival and show me
a picture of a lively lantern display.
Can you describe an autumn scene with vivid orange-red leaves under a
clear blue sky? Then draw an image of it.
What are some interesting behaviors of cats? Can you show me a picture of
 a cat marking its territory?
Can you quickly draw a picture of the Christ the Redeemer statue in Rio
de Janeiro and share some interesting facts about it?
```

```
Hey! Can you describe a spring garden scene? I really want to hear enough
 details, and you need to draw it according to the description!
Do you know what happened 100 years ago? Please draw a history-related
picture! Thank you!
I need a picture of the age of the dinosaurs now. Do you know about past
history? Please draw a picture for me.
Can you describe what a basketball court is like? Draw a basketball
moment for me.
What is the Lantern Festival like? Can you show me some pictures of
traditional lanterns?
What kind of casual outfit do you think is suitable for wearing on a
relaxed Saturday afternoon? Can you draw what it looks like?
Can you describe and perhaps draw a picture showing a person practicing
yoga in a tranquil park at sunrise?
I need a landscape picture of the countryside. Please describe it and
draw an image for me.
Can you draw a picture of an airplane for me? Also, give me some popular
science knowledge about it.
...
```

## I  HUMAN ANNOTATION PLATFORM

We develop a human annotation platform to evaluate the quality of interleaved image-text responses. Annotators assess each response across four predefined dimensions, focusing on the content and coherence between visual and textual elements. To ensure annotation reliability, cross-validation is conducted on high-rated samples. An overview of the annotation interface is shown in Figure 8.

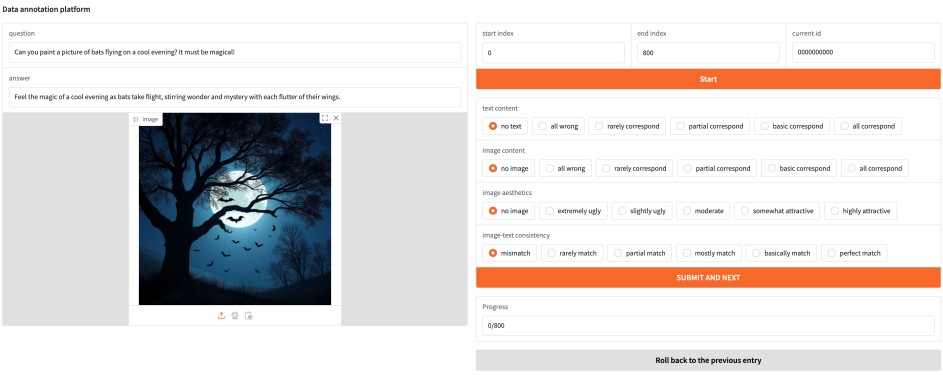

Figure 8: Human annotation platform to evaluate the quality of interleaved image-text responses.

## J  LIMITATIONS OF THIS STUDY

While our work introduces InterSyn, the first large scale, instruction-following dataset for multi-turn, interleaved image-text dialogues, and proposes SynJudge, a comprehensive automatic evaluator emphasizing image-text synergy, several limitations remain that suggest directions for future improvement.

First, although our SEIR framework substantially enhances output quality through multi-stage refinement, the visual fidelity of generated images is inherently constrained by the upper bounds of current text-to-image models. This may limit the expressiveness and precision of visual responses, particularly for fine-grained or specialized topics.

Second, our current dataset is restricted to one image per dialogue turn, which simplifies the modeling process but diverges from real-world scenarios where understanding or generating multiple images simultaneously is often necessary—e.g., comparative reasoning, procedural steps, or spatial reasoning tasks. While we have experimentally validated the feasibility of multi-image dialogue

generation using alternative synthesis pipelines, such functionality is not yet reflected in the released dataset.

Third, the SynJudge evaluator is currently designed to assess single-image responses, meaning it does not fully capture the additional complexity and multimodal dependencies introduced by multi-image contexts. Extending SynJudge to support multi-image evaluation is a promising future direction.

Finally, although InterSyn spans diverse domains and fine-grained topics, future work could enhance its coverage of highly structured tasks or multi-modal reasoning chains that involve deeper world knowledge or long-term dialogue coherence.

These limitations highlight important opportunities for scaling interleaved image-text datasets and improving evaluators toward more generalizable, high-fidelity multimodal generation systems.

