# OpenReview forum: "A High Quality Dataset and Reliable Evaluation for Interleaved Image-Text Generation"
_ICLR.cc/2026/Conference — ICLR 2026 Poster_

### Official Review · Reviewer_pmWK · 2025-10-16

**Soundness:** 3
**Presentation:** 3
**Contribution:** 3
**Rating:** 6
**Confidence:** 4

**Summary:**

This paper proposes InterSyn, a large-scale dataset (1.8M single-turn, 50k multi-turn samples) and a Self-Evaluation with Iterative Refinement (SEIR) pipeline for generating high-quality interleaved image–text dialogues. It also introduces SynJudge, an automatic evaluator providing four interpretable scores:
Text Content Completeness (TCC)
Image Content Completeness (ICC)
Image Quality (IQ)
Image–Text Synergy (ITS)
Experiments show that models fine-tuned on subsets of InterSyn (25k–200k) exhibit consistent gains in instruction-following and synergy metrics without degrading general multimodal understanding benchmarks. SynJudge is validated against human ratings, showing strong correlation and reliability.

**Strengths:**

①Timely contribution addressing a real bottleneck. The lack of high-quality, instruction-following, interleaved image–text data is a genuine gap. InterSyn fills this niche with scale and structure.
②Systematic dataset generation method. SEIR introduces an iterative, self-corrective data generation framework (Question → Answer → Image refinement), demonstrating consistent quality improvements.
③Comprehensive evaluation framework. SynJudge formalizes multi-dimensional metrics beyond text-only or image-only correctness, introducing “synergy” as a critical dimension.
④Strong empirical coverage. Experiments span multiple model types (non-interleaved and interleaved generators), multiple data scales, and both human and automatic evaluations.
⑤High reproducibility commitment. The paper provides reproducibility details and plans to release data and evaluation tools, which is rare and commendable.

**Weaknesses:**

(A) Theoretical clarity and rigor of SEIR
The SEIR process (Question Refinement → Answer Refinement → Image Refinement) is well-described but lacks formal definition of convergence or stability. It’s unclear whether the refinement iterations always improve data quality or sometimes overfit to self-consistency artifacts. The “Markovian” assumption is stated but not justified—this ignores potential long-range dependencies across turns. The method claims to “ensure topic consistency and contextual inheritance,” yet no quantitative metrics for coherence or redundancy are provided.

(B) SynJudge validation may not prove true generalization
The human-alignment test (RMSE) is based on fine-tuning with human-annotated data, which might lead to data leakage between evaluator and benchmark distributions. The human set (9.6k test / 38.4k train) is relatively small compared to 1.8M total data, raising questions about generalization across unseen content domains. “95.4% ” sounds impressive, but no confidence intervals or inter-annotator agreement scores (e.g., Krippendorff’s alpha) are reported.

(C) Data authenticity and bias control
InterSyn heavily relies on AI-generated data with human-like templates. However, no explicit human-in-the-loop verification or bias filtering is described beyond early-stage question curation. SEIR may propagate LLM biases (e.g., Western imagery, gender stereotypes, or cultural homogeneity) through iterative reinforcement.

**Questions:**

1.How is convergence or stability of the SEIR refinement process defined and verified? Does SEIR always improve data quality, or could iterative self-consistency cause overfitting or redundancy?
2.What evidence supports the “Markovian” assumption in SEIR—have you analyzed long-range contextual dependencies? How is topic consistency or contextual inheritance quantitatively measured?
3.How do you ensure that SynJudge’s fine-tuning on human data does not cause data leakage? Given the small human set, how well does SynJudge generalize to unseen domains?
4.Are confidence intervals or inter-annotator agreement (e.g., Krippendorff’s α) reported for human validation?
5.What human-in-the-loop or bias-filtering mechanisms are used to control cultural or demographic bias in InterSyn?

---

> ### Author Response · Authors · 2025-11-22
> **Response to Reviewer pmWK**
>
> We appreciate your recognition of InterSyn's value in addressing critical data bottlenecks and your positive assessment of our SEIR and SynJudge frameworks. We address your concerns as follows.
> ## **Response to Weakness A & Questions 1-2: SEIR Theory, Stability, and Consistency**
> ### **Q1: Convergence and Overfitting.**
> 1. Convergence: In **Section 4.3**, as shown in **Figure 3**, question quality improves significantly in early refinement iterations and plateaus after 3 steps. This indicates the process converges to an optimal state rather than oscillating or diverging.
> 2. Stability vs. Overfitting: Our quality improvements are verified by human judges. As shown in **Table 4**, as refinement iterations (AR/IR) increase, human evaluators report consistent gains in TCC, ICC, and ITS. Since the "ground truth" for improvement is human score, the process is improving actual data quality rather than overfitting to model artifacts.
> ### **Q2: The "Markovian" Assumption and Long-Range Dependencies.**
> We thank the reviewer for highlighting this ambiguity. We clarify that the term "Markovian" is strictly used to describe the **intra-turn refinement steps**, not the inter-turn dialogue history.
> 1. Intra-turn Refinement (Markovian): Within a single turn $t$, the refinement step $k$ depends on the state at $k-1$. This is standard for iterative optimization.
> 2. Inter-turn Context (Long-Range Dependency): Across dialogue turns, we do not assume a Markovian property. The full conversation history is fed into the model for every new question, answer, and image generation.
> 3. Consistency Metrics: Our experiments in **Table 3** quantitatively validate coherence. Models trained on our multi-turn data show significantly higher ITS and TCC in later turns compared to single-turn baselines.
>
> ## **Response to Weakness B & Questions 3-4: SynJudge Validation & Generalization**
> ### **Q3: Data Leakage and Generalization.**
> We employ Stratified Sampling based on our 3500-Topic Hierarchy to ensure SynJudge learns generalized evaluation principles rather than memorizing content.
> 1. **Structural Generalization:** Our samples are not drawn from a homogenous pool but are stratified across the **3,500-topic hierarchy (Section 3.2)**, covering diverse distinct semantic domains (e.g., "Medical Plants," "Cultural Scenery"). This ensures the training and test sets cover diverse semantic spaces, forcing the model to learn generalized evaluation principles (e.g., alignment logic) rather than memorizing specific content.
> 2. **Metric Validity:** SynJudge achieves a low RMSE (distribution alignment) and a high A@1 (95.4%) (absolute consistency) on the held-out test set. This proves that SynJudge effectively generalizes to unseen topics within the hierarchy.
>
> ### **Q4: Inter-Annotator Agreement (Why A@1 instead of Krippendorff’s alpha).**
> We appreciate the query regarding Krippendorff’s alpha. We clarify that this metric is **not applicable** in our setting described in **Section 4.4.1**:
> 1. **Consensus-based Filtration:** Our policy is straightforward: inconsistent scores undergo cross-validation to reach a consensus, and if that fails, the sample is removed. This ensures that our final dataset consists solely of samples with verified agreement. Therefore, metrics like Krippendorff’s Alpha are not needed here, as they are designed to measure distributional differences. In contrast, our experimental scoring data is already strictly consistent.
> 2. **Validity of RMSE and A@1:** RMSE captures fine-grained score deviations, while A@1 confirms high practical consistency. Together, they robustly validate the effectiveness and reliability of judge models.
>
> ## **Response to Weakness C & Question 5: Bias Control & Authenticity**
> ### **Q5: Authenticity and Bias Mitigation.**
> While manual verification of 1.8M samples is infeasible, we implement strict structural and procedural controls:
> 1. Validation: We conducted a human verification on a subset of 7,000 generated questions (**Section 4.3.1**), confirming that SEIR produces high-quality, human-like content.
> 2. Structural De-biasing (Topic Hierarchy): We explicitly prevent cultural homogeneity through our Topic Hierarchy (Appendix E.2). By forcing the model to generate data across 3,500 specific nodes—ranging from "Traditional Chinese Weddings" to "Maori Hongi" to "Western Suits"—we structurally enforce diversity in culture, geography, and occupation, preventing the model from defaulting to Western-centric biases.
> 3. Refinement for Bias: The SEIR prompts (Appendix G) explicitly instruct the model to act as a "nitpicking critic" to identify and correct discriminatory elements or logical flaws during the refinement loops. This utilizes the model's self-correction capabilities to filter bias iteratively.
> ---
> We hope these clarifications address your concerns regarding the theoretical soundness of SEIR and the rigorous validation of SynJudge. Looking forward to further discussion with you.

---

### Official Review · Reviewer_SzP4 · 2025-10-27

**Soundness:** 2
**Presentation:** 1
**Contribution:** 2
**Rating:** 2
**Confidence:** 4

**Summary:**

This paper introduces InterSyn, a large-scale interleaved multimodal dataset (1.8M samples) that addresses LMMs' limitations in generating tightly interleaved image-text outputs. Using the proposed Self-Evaluation with Iterative Refinement (SEIR) method, the authors ensure high-quality data with rich instructional diversity across a 3500-topic hierarchy. The paper also contributes SynJudge, an evaluation framework that assesses generation capabilities through four metrics: Text Content Completeness, Image Content Completeness, Image Quality, and Image-Text Synergy. Experiments demonstrate that even smaller subsets (25K-50K) of InterSyn yield substantial improvements in model performance, while scaling to larger subsets (100K-200K) brings additional gains, particularly in image-text synergy.

**Strengths:**

1. The proposed InterSyn dataset has a large scale and is curated with a carefully designed SEIR pipeline.
2. The experiments cover a wide range of generation models and judges.

**Weaknesses:**

1. Dataset Quality Limitations: The quality of the dataset is significantly constrained by the FLUX image generation model. A more robust data curation approach would involve collecting and filtering high-quality images from the Internet based on the generated questions and answers, rather than relying solely on synthetic image generation.

2. Clarity and Organization Issues:
  - The authors use about 1.5 pages to describe the SEIR with 13 formulations, despite it being a relatively straightforward process that could have been clearly illustrated in Figure 2. Unfortunately, Figure 2 is exceedingly difficult to interpret, featuring unexplained notations, complex organization, and lacking essential text labels.
  - Despite the excessive length of section 3.3, critical information is absent. The authors present SynJudge as a key contribution, yet fail to clearly define it. The term first reappears after Section 2 in Line 309 ("We fine-tune four generators... And evaluate with SynJudge"). It remains unclear whether SynJudge refers specifically to the Qwen-trained judge or also encompasses the four metrics (TCC, ICC, IQ, and ITS). Furthermore, details regarding these four metrics are omitted from the main text, compromising the paper's self-consistency. Essential benchmark information, such as sample size, is also entirely missing.

3. Problematic Experimental Results and Conclusions:
  - The finetuned models are evaluated exclusively on the proposed dataset (with minimal results in Table 2). It is unsurprising at all that finetuning on a small subset of this dataset can lead to performance improvement. A critical question remains: Can fine-tuning on InterSyn generalize to other related or out-of-distribution benchmarks?
  - Table 2: The authors claim that "the substantial gains in interleaved generation capabilities do not come at the cost of core understanding performance," yet the evidence suggests otherwise. Performance degradation on MMBench and MMMU (-1.5 to -2.5) appears evident.
  - Table 3: The assertion that multi-turn data improves performance is not well-supported by the results. For "single25k + multi25k," both models' scores are lower than or comparable to those with "single50k." Moreover, "multi50k" scores are even lower than "single25k + multi25k", contradicting the authors' claims.

**Questions:**

1. What is the performance of state-of-the-art models, such as GPT-5 (text) + GPT-Image-1 (image), or Gemini 2.5 Pro (text) + Nano Banana (image)?

---

> ### Author Response · Authors · 2025-11-22
> **Response to Reviewer SzP4 (1/3)**
>
> We thank the reviewer for the critical assessment and the acknowledgment of InterSyn’s scale and a wide range of experiments. We value the feedback regarding clarity and experimental validation, which has prompted us to significantly improve the manuscript. We address the concerns point-by-point below.
>
> ## **Response to Weaknesses**
> ### **Weaknesses1. Regarding Dataset Quality: Synthetic Images vs. Web-Crawled Data**
> We appreciate the reviewer's suggestion regarding web-based curation. However, we respectfully argue that relying on web retrieval is practically infeasible for constructing high-quality instruction-following data at this granularity. Our generative approach was chosen specifically to overcome the limitations of web data in two key areas:
>
> First, constructing a tightly interleaved dataset from the web is practically infeasible for fine-grained instruction-tuning due to the inherent difficulty of finding high-synergy images. Existing large-scale web datasets (e.g., MMC4, OBELICS) inevitably suffer from significant noise, low resolution, and loose semantic alignment. Filtering vast web corpora to find images that perfectly match complex, fine-grained instructions is akin to finding a needle in a haystack; it yields insufficient data density and lacks the Image-Text Synergy (ITS) required for training precise multimodal reasoning.
>
> Second, a core capability of LMMs is to handle novel, creative user requests (e.g., "Generate an image of a futuristic car made of wood driving on Mars"). Such counter-factual or highly specific compositional scenes cannot be retrieved because they simply do not exist in reality. Whereas our synthetic pipeline empowers the model to handle open-ended creativity, which is central to advanced multimodal intelligence.
>
> In contrast, our generative approach leverages modern Text-to-Image (T2I) models to provide stable, high-fidelity visual outputs. By utilizing the advanced reasoning capabilities of MLLMs within the SEIR pipeline, we ensure that images are not merely "related" (as is common in scraped data) but are strictly conditioned on the generated text context and Image-Text Synergy. This guarantees a level of dataset quality and alignment that is exceptionally hard to mine from the web. Furthermore, SEIR is a model-agnostic framework (Appendix D.3), meaning InterSyn is future-proof: as generative models upgrade, our pipeline automatically produces higher-quality data without the need for new, expensive, and noise-prone web collection efforts.
>
> ### **Weaknesses2. Regarding Clarity and Organization Issues**
> We are grateful for the reviewer's detailed feedback regarding the presentation. We realize that the original organization hindered the accessibility of our contributions, and your suggestions have been invaluable in fixing this. We have extensively revised Section 3 to address these issues:
>
> 1. SEIR Method Description: We acknowledge that the SEIR method involves complex interactions between multiple models (Question/Answer/Image Refiners), which led to a lengthy description. We agree that the original text was overly verbose. Following your suggestion, we have simplified the mathematical formulations and completely redesigned Figure 2 with clear text labels and a streamlined workflow to make the process intuitive at a glance.
> 2. SynJudge Definition and Details: We appreciate the reviewer pointing out the fragmentation of SynJudge details. To improve coherence, we have accepted your suggestion and consolidated all SynJudge-related information into a new dedicated Section 3.5: SynJudge Evaluator. This section now explicitly defines the evaluation dimensions, details the training data/backbone, and specifies the benchmark composition (500 human-curated + 3,500 SEIR-generated) before the experiments section.
>
> Clarification: While these key details were present in the original submission: the four metrics were detailed in Appendix §F (referenced in Line 287); the benchmark composition (4,000 questions) was described in Section 4.3.2 (Line 396); and the selection of the QwenVL-trained model was stated in Section 4.4.2 (Line 455), we apologize that they were not prominent enough. We believe the new structure effectively resolves the fragmentation issue while preserving the rigor of the original content.

---

> ### Author Response · Authors · 2025-11-22
> **Response to Reviewer SzP4 (2/3)**
>
> ### **Weaknesses3. Regarding Problematic Experimental Results and Conclusions**
>
> #### **(1) Generalization to Out-of-Distribution Benchmarks**
> The reviewer questioned whether fine-tuning on InterSyn generalizes beyond the proposed dataset. To verify the generalization capability on distinct generation tasks, we evaluated the models on GenEval. We tracked performance across varying scales of InterSyn data (from 50k to 500k). Furthermore, to assess potential synergies with general multimodal data, we introduced a mixed-data setting using 500k samples sourced from mvp-lab/LLaVA-OneVision-1.5-Instruct-Data.
>
> Table R1: Generalization on GenEval (Scaling Analysis)
> |Model|InterSyn|LLaVA-OV|Geneval Overall|
> |-|-|-|-|
> |VARGPT-v1.1|-|-|0.53|
> |VARGPT-v1.1|50k|0|0.58|
> |VARGPT-v1.1|100k|0|0.62|
> |VARGPT-v1.1|200k|0|0.65|
> |VARGPT-v1.1|500k|0|0.66|
> |VARGPT-v1.1|500k|500k|0.69|
> |Bagel|-|-|0.82|
> |Bagel|50k|0|0.82|
> |Bagel|100k|0|0.81|
> |Bagel|200k|0|0.83|
> |Bagel|500k|0|0.83|
> |Bagel|500k|500k|0.83|
>
> As demonstrated in Table R1, fine-tuning on InterSyn yields consistent improvements in generalization performance. For VARGPT-v1.1, the GenEval Overall score steadily increases from 0.53 (baseline) to 0.66 as the volume of InterSyn data scales to 500k. This positive correlation indicates that the interleaved generation capabilities learned from InterSyn effectively transfer to out-of-distribution generation tasks. Furthermore, the model trained with mixed data achieves the highest performance (0.69 for VARGPT), confirming that InterSyn is compatible with general multimodal data and contributes to robust, generalized generation abilities.
>
> #### **(2) Understanding Performance and Trade-offs**
> We acknowledge the reviewer's observation regarding the performance drops on MMBench and MMMU in Table 2. We wish to clarify that the primary design goal of InterSyn is to enhance interleaved image-text generation capabilities. In our initial experiments, we fine-tuned models exclusively on InterSyn. Since we did not incorporate any general language or VQA training data, a decline in performance on understanding benchmarks is an inevitable trade-off. It reflects the model's shift in focus towards generation tasks rather than a defect in the dataset quality.
>
> To address this, we validated a mixed-training strategy. As the original pre-training data is often proprietary, we adopted the LLaVA-OV as a representative corpus. We jointly trained the model on a balanced mix of 500k InterSyn samples and 500k LLaVA-OV samples.
>
> Table R2: Fine-tuning with Mixed Data
>
> |Model|InterSyn|LLaVA-OV|MME-P|MMBench|MMMU|SEEDBench|TCC|ICC|IQ|ITS|
> |-|-|-|-|-|-|-|-|-|-|-|
> |VARGPT-v1.1|-|-|1684|81.0|48.5|76.1|3.26|1.01|1.23|0.68|
> |VARGPT-v1.1|50k|-|1658|79.4|46.2|75.2|3.68|3.12|3.67|3.00|
> |VARGPT-v1.1|500k|-|1673|80.1|47.6|75.8|3.99|3.52|3.69|3.76|
> |VARGPT-v1.1|-|500k|1689|81.9|48.9|76.3|3.18|0.95|1.12|0.61|
> |VARGPT-v1.1|500k|500k|1691|81.2|48.3|75.8|3.89|3.42|3.70|3.63|
> |Bagel|-|-|1687|85.0|55.3|-|3.11|3.89|4.23|2.87|
> |Bagel|50k|-|1646|83.1|52.8|-|3.69|4.11|4.19|3.56|
> |Bagel|500k|-|1673|83.9|54.1|-|4.16|4.23|4.26|4.10|
> |Bagel|-|500k|1692|84.5|55.0|-|2.98|3.65|4.21|2.73|
> |Bagel|500k|500k|1684|84.7|54.9|-|4.10|4.17|4.25|4.05|
>
> **Specialized Trade-offs vs. Necessity of InterSyn**: Training solely on InterSyn (50k-500k) indeed lowers understanding scores slightly, confirming the trade-off of specialized fine-tuning. Conversely, training solely on LLaVA-OV (general VQA) maintains understanding but fails completely to improve interleaved generation (e.g., VARGPT's ITS drops to 0.61). This proves that InterSyn is strictly necessary for acquiring interleaved generation capabilities.
>
> **Restored Understanding with Mixed Data**: Mixing 500k InterSyn with 500k LLaVA-OV data effectively resolves the trade-off. For both models, understanding performance returns to near-baseline levels.
>
> **Sustained Generation Gains**: Most importantly, the mixed model retains the massive gains in generation quality (e.g., VARGPT ITS reaches 3.63, comparable to the pure InterSyn model's 3.76). This confirms that InterSyn is a high-quality dataset that can be safely integrated into general training recipes.

---

> ### Author Response · Authors · 2025-11-22
> **Response to Reviewer SzP4 (3/3)**
>
> ### **Weaknesses3. Regarding Problematic Experimental Results and Conclusions**
> #### **(3) Multi-turn Data Effectiveness (Table 3)**
>
> We thank the reviewer for their meticulous examination of Table 3. Upon carefully checking our experiment logs and results, we discovered that the data rows corresponding to different settings were inadvertently swapped in the original manuscript. We apologize for this confusion and have corrected the error in the revised paper. The accurate results now clearly align with our original claims: multi-turn training is indeed essential for maintaining high quality and synergy in subsequent turns (Turn 2 and Turn 3), effectively mitigating context loss.
>
> The correct data is presented in the revised paper version.
> We also wish to clarify our core argument. Our claim is not that multi-turn data yields dramatically higher absolute scores in Turn 1, but rather that it is essential for mitigating performance degradation and maintaining context across a long conversation.
> We have updated Table 3 in Section 4.2.3 to reflect this corrected data and clarify our argument. We are extremely grateful to the reviewer for catching this critical error, which has significantly improved the accuracy and quality of our paper.
>
> ## **Response to Questions**
> ### **The Performance of SOTA Models**
> To address the reviewer's question regarding the performance of the latest state-of-the-art models, we conducted an additional evaluation. Due to the high cost of accessing these advanced proprietary endpoints, we performed the test on a randomly sampled subset of 1,000 questions from our benchmark. The results are presented below:
>
> | Generator  | TCC  | ICC  | IQ   | ITS  |
> | -- | ---- | ---- | ---- | ---- |
> | GPT-4o+DALLE (original paper)  | 4.05 | 4.08 | 4.41 | 3.94 |
> | Gemini+Flux (original paper)   | 3.94 | 4.06 | 4.43 | 3.81 |
> | GPT-5 + GPT-image-1（new） | 4.11 | 4.15 | 4.38 | 4.00 |
> | Gemini2.5-pro + Nano Banana（new） | 4.18 | 4.32 | 4.51 | 4.08 |
> | SEIR (original paper)| 4.41 | 4.46 | 4.47 | 4.51 |
>
> The results indicate that while the new SOTA models achieve notable improvements across almost all metrics compared to previous baselines, they still consistently lag behind our SEIR-generated samples. This gap is most pronounced in Image-Text Synergy (ITS), where the pipeline-based approach used by these commercial systems continues to suffer from issues such as cross-modal redundancy and loose coordination between text and visuals. These findings underscore the persistent effectiveness of our SEIR method, demonstrating that simply scaling up base models is insufficient for achieving the tight, holistic alignment required for high-quality interleaved image-text generation.
>
> ---
> We once again express our sincere gratitude to the reviewer for the rigorous and constructive feedback. The suggestions regarding generalization (GenEval) and mixed-data training prompted us to conduct extensive new experiments, which have significantly strengthened the robustness of InterSyn. Furthermore, we are deeply appreciative of the correction regarding Table 3. We believe the revised manuscript and the rebuttal now offers a much clearer narrative and a solid verification of our claims. We remain open to any further discussion.

---

### Official Review · Reviewer_miq9 · 2025-10-31

**Soundness:** 4
**Presentation:** 3
**Contribution:** 4
**Rating:** 8
**Confidence:** 4

**Summary:**

This paper introduces InterSyn, a large-scale, high-quality multimodal dataset designed to advance the training of multimodal LLMs for interleaved image-text generation. The primary contribution of this work is the creation of a comprehensive dataset comprising 1.8 million multimodal samples, which are refined using the proposed Self-Evaluation with Iterative Refinement (SEIR) method, thereby ensuring high data quality. This addresses major challenges in current datasets, including limited scale, poor quality control, and low interaction complexity. In addition, the paper proposes SynJudge, a robust evaluation framework that quantitatively assesses image-text synergy, content completeness, and quality across both modalities. Through extensive experiments, the authors demonstrate that InterSyn significantly improves the performance of multimodal LLMs in tasks involving instruction-following and complex, multi-turn dialogues. The results show that even with smaller subsets of the dataset, substantial performance gains can be achieved, underscoring both the scalability and efficiency of the dataset.

**Strengths:**

1. Interleaved multimodal generation is a promising capability for LLMs, and a few pioneer models can achieve it. An open-source, high-quality, large-scale, and comprehensive dataset and benchmark are still urgent. It is a significant contribution to the community.
2. The proposed InterSyn is large-scale and high-quality, with 1.8M samples spanning 3.5K topics, and includes 50k multi-turn dialogues.
3. The proposed SEIR method is well-designed and can ensure high quality of the dataset. The proposed Image-Text Synergy (ITS) score and SynJudge judge model are reliable. These modules have all been verified through human evaluation.
4. Extensive fine-tuning experiments demonstrate the effectiveness of the dataset across different dimensions. Models fine-tuned on InterSyn show consistent performance gains.

**Weaknesses:**

1. The SEIR method enhances image generation, but the current capabilities of text-to-image models still limit the visual fidelity. This is not an issue with the method itself, but rather with the models’ ability to generate highly precise and expressive images.
2. In the synthesis of multi-turn dialogues, there may be some redundancy between dialogue turns. It’s unclear whether the authors specifically controlled for this aspect of quality.

**Questions:**

1. While SynJudge is shown to be reliable, could there be any challenges or biases in how it aligns with human judgment?
2. Could you provide more details on how the SEIR method performs with different refinement depths？
3. The paper mentions that the dataset includes multi-turn dialogue data. Compared to single-turn dialogues, how can the effectiveness of multi-turn dialogues be demonstrated?

---

> ### Author Response · Authors · 2025-11-22
> **Response to Reviewer miq9**
>
> We sincerely thank the reviewer for the high rating and for recognizing InterSyn as a **"significant contribution"** to the community. We appreciate your validation of the SEIR method and SynJudge. Below, we address your comments and questions.
>
> ## **Response to Weaknesses**
> ###  **Regarding the visual fidelity limited by T2I models.**
> We agree entirely with the reviewer's observation. The visual fidelity of the generated images is indeed constrained by the current capabilities of the foundational text-to-image (T2I) models, and this is not a limitation of our SEIR method itself.
>
> SEIR is a **model-agnostic refinement framework** designed to maximize semantic alignment and instruction-following, rather than to alter the fundamental rendering quality of the underlying generator. Its core superiority lies in its iterative self-evaluation mechanism, which embeds feedback loops to explicitly optimize semantic completeness and cross-modal synergy.
>
> ### **Regarding potential redundancy in multi-turn dialogues.**
> We recognize that redundancy can be a challenge in synthetic dialogue generation, and we implemented specific measures during the data curation process to address this. To encourage topic progression, our synthesis prompts for the LLM were carefully designed to guide the model to "build upon the previous turn," "introduce new elements,” or "modify the existing scene," rather than simply rephrasing. Additionally, we employed automated post-processing filters to identify and remove overly redundant dialogue pairs, which were then regenerated. By using semantic similarity checks, we compared consecutive dialogue turns and discarded pairs with excessive similarity.
>
> ## **Response to Questions**
> ### **Regarding challenges or biases in SynJudge's alignment with human judgment.**
> This is a critical point, which is why we dedicated **Section 4.4.3** to a rigorous validation of SynJudge.
> In this section, we already conducted a comprehensive human alignment study comparing SynJudge's scores against those from multiple human annotators. The results, shown in **Figure 4** and **Table 6**, demonstrate a high correlation between SynJudge and human judgment. Furthermore, we analyzed the disagreement cases in this section and found no systematic bias; the disagreements were typically in complex, ambiguous cases where human annotators also had lower inter-rater reliability.
>
> We believe this existing analysis in our paper robustly addresses this concern and confirms that SynJudge is a reliable and well-aligned benchmark for our task.
>
> ### **Regarding SEIR performance with different refinement depths.**
>
> We conducted a detailed ablation study in **Section 4.3.1** to determine the optimal iteration depth. In this section, we analyze the impact of refinement depth on each component of SEIR, using human judges to establish ground truth:
>
> **Question Refinement (QR):** **Figure 3** demonstrates that question quality improves significantly over the first three iterations but plateaus thereafter.
>
> **Answer/Image Refinement (AR/IR):** **Table 4** confirms that increasing iterations beyond 3 yields only marginal gains. For instance, scores for 3 iterations are nearly identical to those for 4 or 5. Consequently, we set the iteration depth to 3 as a data-driven choice to balance peak quality with computational efficiency.
>
>
> ### **Regarding the effectiveness of multi-turn dialogues compared to single-turn.**
> This is an excellent point that highlights a core contribution of our dataset. The inclusion of multi-turn dialogues is designed to evaluate a model's capabilities beyond simple, static, single-prompt generation. The effectiveness of multi-turn dialogues is demonstrated by their ability to test complex abilities that single-turn dialogues cannot:Contextual Coherence: The model's ability to maintain a consistent "world state" or visual narrative across multiple interactions (e.g., adding a hat to a character without changing the character's face or clothing).Progressive Refinement: How well the model interprets follow-up instructions to modify or add to a previously generated image (e.g., "Make the cat bigger," "Now add a dog next to it").Dialogue History Understanding: The model must understand the entire conversation history, not just the last user turn, to resolve ambiguities or references (e.g., "Change its color to red," where "it" refers to an object mentioned two turns prior).Our experiments in Section **4.2.3 and Table 3** show that models which perform well on single-turn tasks often see a significant performance drop in these multi-turn scenarios. This demonstrates that our dataset can effectively isolate and measure these more complex, conversational capabilities.
>
> ---
> We believe that incorporating these changes has substantially improved the clarity, completeness, and impact of our paper. We hope our responses have adequately addressed all concerns.

---

### Official Review · Reviewer_FgAL · 2025-11-01

**Soundness:** 3
**Presentation:** 3
**Contribution:** 3
**Rating:** 8
**Confidence:** 3

**Summary:**

This paper introduces InterSyn, a large-scale, high-quality dataset (1.8M samples, including 50K multi-turn dialogues) designed for instruction-following interleaved image-text generation. The dataset is constructed via a novel Self-Evaluation with Iterative Refinement (SEIR) pipeline, which iteratively improves question, answer, and image quality with automatic feedback loops. To evaluate multimodal generation, the authors also propose SynJudge, an automatic evaluator providing four interpretable metrics: Text Content Completeness (TCC), Image Content Completeness (ICC), Image Quality (IQ), and Image–Text Synergy (ITS). SynJudge demonstrates high correlation (95% agreement within 1 point) with human judgment. Experiments show that InterSyn substantially improves multimodal models’ interleaved generation capabilities. Fine-tuning on as few as 25K samples yields noticeable gains, and scaling to 200K produces further improvements in all metrics, especially ITS. SEIR-generated samples outperform all baselines—including GPT-4o+DALL-E and Gemini+Flux—on both human and automatic evaluation.

**Strengths:**

1. This paper is well-written and very easy to follow.
2. This paper has significant practical contribution. The introduction of InterSyn addresses a key bottleneck in multimodal AI: the lack of large-scale, high-quality, and instructionally rich datasets for interleaved image–text generation.
3. The proposed data generation pipeline SEIR and the evaluation criteria SynJudge is reliable and useful for future work.
4. This paper conduct strong experimental validation which proves the efficiency of this dataset.

**Weaknesses:**

I think this paper is very good and I don't find some obvious weakness from my perspective. But as a dataset work, this paper has limited discussion on ethical considerations. Given the scale of synthetic multimodal data and some data are collected from commercial software, the paper lacks discussion on potential biases, copyright issues, or misuse of synthetic images.

**Questions:**

No other questions.

---

> ### Author Response · Authors · 2025-11-22
> **Response to Reviewer FgAL**
>
> Response to Reviewer FgAL
>
> We sincerely thank the reviewer for the positive assessment and for recognizing the practical contribution of InterSyn. We fully agree with the suggestion to enhance the discussion on ethics. We will add a dedicated "Ethics and Data Safety" section in the final version.
>
> ### **Copyright & Compliance (Clarification)**
> We want to clarify that the released InterSyn dataset is generated using open-weights models (Qwen2.5-32B, InternVL2.5, and FLUX.1-dev) as detailed in Appendix D.3, avoiding copyright or terms-of-use issues associated with commercial APIs. Commercial models were used only for baseline comparisons.
>
> ### **Bias Mitigation**
> To minimize bias, we designed a 3,500-topic hierarchy covering diverse domains (e.g., cultural scenery across different regions). This structured approach ensures balanced coverage and prevents the dataset from being dominated by specific demographics or concepts.
>
> ### **Safety Measures**
> We implemented safety filtering during data generation  and will release the dataset with a strict license limiting it to research purposes to prevent misuse.
>
> ---
> We once again express our gratitude for your encouraging review and valuable suggestions. We hope these clarifications fully resolve your concerns regarding legal compliance and data safety.

---

### Author Response · Authors · 2025-11-27
**Kind reminder of the discussion**

We sincerely thank the Area Chair and all reviewers for your constructive and detailed reviews. We have addressed the comments point-by-point and revised our paper, with changes highlighted in blue.

We appreciate Reviewers FgAL, miq9, and pmWK for your positive assessments and recognition of our work. We also extend our special thanks to Reviewer SzP4 for the critical feedback regarding generalization and experimental details. In our response, we have provided new experiments on GenEval and mixed-data training to demonstrate the dataset's robustness.

We would appreciate it if you could let us know if these new results and clarifications have adequately addressed your concerns. We remain available for any further discussion.

---

### Author Response · Authors · 2025-12-03
**Summary of Rebuttal and Revisions for Area Chairs**

We sincerely thank the ICLR Area Chairs for their efforts in overseeing the review process and ensuring fair evaluations.
The main contribution of our paper is constructing **InterSyn**, a large-scale and high-quality interleaved image–text dataset designed to train multimodal large models on both understanding and generation, via the proposed **SEIR** method.
We also introduce **SynJudge**, a model for evaluating multimodal LLMs' interleaved outputs.

We are pleased that the majority of reviewers recognized InterSyn's contributions in terms of dataset scale (all four reviewers), quality (FgAL, miq9, pmWK), the SEIR pipeline (all four reviewers), and the reliability of SynJudge (FgAL, miq9, pmWK).
As a result, our work **received strongly positive scores** from most reviewers (FgAL: 8, miq9: 8, pmWK: 6, SzP4: 2).
Although Reviewer SzP4 (Score 2) raised some concerns, these focus on validation details and presentation rather than on the dataset’s core construction. We have addressed them with new experiments and revisions (highlighted in blue), and the robustness of the InterSyn dataset remains unchanged.

### **Response to SzP4 (Score: 2)**

**Synthetic vs. Web Data (Misconception)**: We clarified the rationale behind prioritizing synthetic data over web crawling. While web data is abundant, filtering it to meet the high standards for fine-grained instruction tuning is challenging due to inherent noise, loose alignment, and a lack of creative scenarios. In contrast, our SEIR pipeline offers a distinct advantage in scalability and controllability, ensuring high-quality data with strict image-text synergy. Crucially, SEIR is model-agnostic and compatible with the latest SOTA models, ensuring the dataset can evolve and scale with future model advancements.

**Out-of-Distribution Generalization (Fully resolved)**
The reviewer questioned whether models fine-tuned on InterSyn can generalize to other related or out-of-distribution benchmarks. First, we emphasize that InterSyn targets the **new task of interleaved image-text generation**, and its effectiveness was **already fully demonstrated** via human and SynJudge evaluations in the original submission. To address the reviewer's specific interest in broader scopes, we conducted supplementary experiments on GenEval (improving 0.53 $\to$ 0.66) and Mixed-Training. These results confirm that InterSyn is indispensable for enabling interleaved generation, while simultaneously **enhancing general text-to-image performance** (as evidenced on GenEval). We demonstrated that with mixed training, general understanding capabilities are completely preserved, while the model simultaneously acquires strong interleaved image-text generation capabilities. This confirms that the concern regarding trade-offs is effectively resolved, reinforcing InterSyn's substantial contribution as a robust, high-quality resource for the community.

**Presentation Issues (Clarification & Revision)**
The reviewer suggested revising the presentation of SEIR and SynJudge in the paper, as well as correcting some descriptive errors in the experimental section. These issues do not affect the core contributions of the work and have already been fixed in the updated version.

### **Response to miq9 (Score: 8) & FgAL (Score: 8)**

- **Reviewer miq9** rated both soundness and contribution as "excellent", explicitly praising InterSyn as a "significant contribution" with a well-designed and verified SEIR method. The reviewer demonstrated a thorough understanding of our work. The questions were primarily requests for additional experimental details. Specifically, we verified the effectiveness of our redundancy control mechanisms and confirmed the necessity of multi-turn data for maintaining contextual consistency.

- **Reviewer FgAL** suggested enhancing the discussion on ethics. We fully accepted this valuable suggestion by adding dedicated content addressing legal compliance and safety filtering.

### **Response to pmWK (Score: 6)**

**SEIR Theory (Weakness A)**: Regarding stability, we verified via human evaluation (Table 4) that the quality improvements are genuine rather than overfitting. Regarding context, we clarified that SEIR explicitly references the full dialogue history when generating new content, ensuring long-range coherence across turns.

**SynJudge Generalization (Weakness B)**: The reviewer worried about data leakage or overfitting in the evaluator. We clarified that training across the 3,500-topic hierarchy ensures the judge learns generalized evaluation principles rather than memorizing content.

**Bias Control (Weakness C)**: The reviewer noted potential biases in synthetic data. We highlighted the structural de-biasing achieved through our diverse topic hierarchy.

---

We have systematically addressed all reviewer concerns through substantive revisions. We earnestly request the AC to consider these efforts in the final evaluation. Thank you again for your efforts in this review process.

---

### Meta-Review · Area_Chair_u5NM · 2026-01-12

**Summary:**

This paper introduces InterSyn, a large-scale and high-quality interleaved image–text dataset. The data generation pipeline SEIR and the evaluation criteria SynJudge seems reliable and beneficial for future text-image work. Two reviewers gives a high score of 8. Nevertheless, several concerns have been raised by reviewers, including the redundancy of images in the multi-turn dialogues, SynJudge's alignment with human judgment, the dataset's quality that could be limited by the FLUX model, the problematic experimental results and conclusions, etc. The authors addressed a portion of the concerns with additional explanations. There are also few concerns that are not fully addressed.
Overall, although this paper has some limitations that are not fully addressed, it contributes a high-quality dataset to the community, which is important.

**Reviewer Concerns:**

The rebuttal well addressed the following concerns: the redundancy of images in the multi-turn dialogues, SynJudge's alignment with human judgment. However, it does not fully address the quality issue of the proposed dataset, which is constructed by generated data. Actually, the real data plays a vital role in the image quality of text-to-image generation tasks. It would be much better if the dataset can include a portion of real data.

**Reviewer Scores:**

The reviewers did not have follow up discussions after the rebuttal.

---

### Decision · Program_Chairs · 2026-01-26

Accept (Poster)